# Functional and structural asymmetry suggest a unifying principle for catalysis in membrane-bound pyrophosphatases

Jannik Strauss [1,6], Craig Wilkinson[1], Keni Vidilaseris [2], Orquidea M de Castro Ribeiro [2], Jianing Liu[2], James Hillier[1,7], Maximilian Wichert [3], Anssi M Malinen[4], Bernadette Gehl[2,8], Lars JC Jeuken [3], Arwen R Pearson[5] & Adrian Goldman [1,2✉]

## Abstract

**Membrane-bound pyrophosphatases (M-PPases) are homodimeric primary ion pumps that couple the transport of $Na^+$- and/or $H^+$ across membranes to the hydrolysis of pyrophosphate. Their role in the virulence of protist pathogens like *Plasmodium falciparum* makes them an intriguing target for structural and functional studies. Here, we show the first structure of a $K^+$-independent M-PPase, asymmetric and time-dependent substrate binding in time-resolved structures of a $K^+$-dependent M-PPase and demonstrate pumping-before-hydrolysis by electrometric studies. We suggest how key residues in helix 12, 13, and the exit channel loops affect ion selectivity and $K^+$-activation due to a complex interplay of residues that are involved in subunit-subunit communication. Our findings not only explain ion selectivity in M-PPases but also why they display half-of-the-sites reactivity. Based on this, we propose, for the first time, a unified model for ion-pumping, hydrolysis, and energy coupling in *all* M-PPases, including those that pump both $Na^+$ and $H^+$.**

**Keywords** Membrane Proteins; Time-resolved Crystallography; Pyrophosphatases; Asymmetry; Mechanism
**Subject Categories** Membranes & Trafficking; Structural Biology

## Introduction

Pyrophosphatases (PPases) catalyse the hydrolysis of inorganic pyrophosphate ($PP_i$), a by-product of nearly 200 biosynthetic reactions across all kingdoms of life (Heinonen, 2001; Lahti, 1983). Soluble PPases (S-PPases) are responsible for recycling the intracellular $PP_i$ pool in all types of organisms, whereas the function of membrane-bound PPases (M-PPases) extends beyond mere $PP_i$ hydrolysis (Baltscheffsky et al, 1966). They utilise the energy stored in the phosphoanhydride bond of $PP_i$ by coupling its hydrolysis to the directed transport of sodium ions ($Na^+$) and/or protons ($H^+$) across membranes but are only present in plants, parasitic protists, and certain prokaryotes (Baltscheffsky et al, 1999; Luoto et al, 2013; Malinen et al, 2007). They are classified into different subclasses based on their ion-pumping selectivity and co-factor dependence. To date, $H^+$-pumping ($H^+$-PPase), $Na^+$-pumping ($Na^+$-PPase) and dual-pumping ($Na^+$,$H^+$-PPase) M-PPases have been found (Luoto et al, 2013; Nordbo et al, 2016), most of which require potassium ($K^+$) for maximal catalytic activity. Only in $H^+$-PPases has evolution given rise to a subclass of $K^+$-independent enzymes (Table 1) (Walker and Leigh, 1981).

Plants, certain prokaryotes and parasitic protists utilise the $PP_i$ pool as an additional energy source to survive low-energy and high-stress conditions by establishing electrochemical gradients across membranes (García-Contreras et al, 2004; Lander et al, 2016). This makes M-PPases a valuable target for structural and functional studies, because they could be of benefit in the fight against existing and emerging challenges to global food security and human health, e.g., by improving the drought tolerance in transgenic plants (Esmaeili et al, 2019; Gaxiola et al, 2001; Park et al, 2005) or by impairing cellular homoeostasis in pathogenic protozoan parasites that harbour M-PPases, such as *Plasmodium* ssp. (malaria), *Leishmania spp.* (leishmaniasis), *Trypanosome spp.* (trypanosomiasis) and *Toxoplasma gondii* (toxoplasmosis) to combat these diseases (Zhang et al, 2018; Lemercier et al, 2002; Liu et al, 2014).

### Structural features of M-PPases

M-PPases are large (66–89 kDa), single-domain integral membrane proteins comprised of two identical monomers, each with 15–17 transmembrane helices (Luoto et al, 2015). To date, the structures of only two are known, a $K^+$-dependent $H^+$-PPase from *Vigna radiata* (*Vr*PPase) and a $K^+$-dependent $Na^+$-PPase from

[1]Astbury Centre for Structural and Molecular Biology, University of Leeds, LS2 9JT Leeds, UK. [2]Molecular and Integrative Biosciences, Biological and Environmental Sciences, University of Helsinki, 00100 Helsinki, Finland. [3]Leiden Institute of Chemistry, University Leiden, PO Box 9502, 2300 RA Leiden, The Netherlands. [4]Department of Life Technologies, University of Turku, FIN-20014 Turku, Finland. [5]Institute for Nanostructure and Solid State Physics, Hamburg Centre for Ultrafast Imaging, Universität Hamburg, 22761 Hamburg, Germany. [6]Present address: Numaferm GmbH, Düsseldorf, Germany. [7]Present address: Bio-Rad Laboratories Ltd., Watford, UK. [8]Present address: Department of Applied Physics, Aalto University, FI-00076, AALTO Espoo, Finland. ✉E-mail: adrian.goldman@helsinki.fi

**Table 1. M-PPase classification into different subclasses.**

| Cation pumping specificity | Monovalent cation dependence | Residue at | | Semi-conserved glutamate | Structural data (PDB) | Example | Reference |
|---|---|---|---|---|---|---|---|
| | | 12.46 | 12.49 | | | | |
| Na+ | K+, Na+ | A | G | 6.53 | 4AV3, 4AV6, 5LZQ, 5LZR, 6QXA, This study | Thermotoga maritima | (Kellosalo et al, 2012; Li et al, 2016; Vidilaseris et al, 2019) |
| H+ | K+ | A | T | 6.57 | 4A01, 5GPI, 6AFS, 6AFT, 6AFU, 6AFV, 6AFW, 6AFX, 6AFY, 6AFZ | Vigna radiata | (Li et al, 2016; Lin et al, 2012; Tsai et al, 2019) |
| | - | K | G | 6.53 | This study | Pyrobaculum aerophilum | - |
| Na+,H+ (dual) | K+, Na+ | A | G | 6.53 | - | Clostridium leptum | - |

*Thermotoga maritima* (*Tm*PPase), with structures available for both in various catalytic states (Kellosalo et al, 2012; Li et al, 2016; Lin et al, 2012; Tsai et al, 2019; Vidilaseris et al, 2019). The helices of each subunit arrange into an inner ring (helices 5, 6, 11, 12, 15, 16) containing the functional core (active site, coupling funnel, ion gate, exit channel) and an outer ring (helices 1–4, 7–10, 13, 14) of largely unknown function (Fig. 1A). In the following, we use the residue numbering scheme $X^{Y.Z}$ in which X represents the amino acid as single-letter code, Y denotes the helix on which it is located and Z defines the offset of a well-conserved residue in the centre of this helix (Ballesteros and Weinstein, 1995). This simplifies residue comparison between proteins and highlights conservation. A translation to conventional residue numbering can be found in Appendix Table S1.

### Active site

The active site in M-PPases contains the completely conserved aspartate, asparagine and lysine residues that provide the basis for $PP_i$ binding and hydrolysis (Kellosalo et al, 2012; Lin et al, 2012). The aspartate and asparagine side-chains coordinate up to five $Mg^{2+}$ ions, which capture $PP_i$ in a metal cage (Fig. 1B), whereas lysine side chains directly stabilise $PP_i$ binding at the active site. Of the $Mg^{2+}$ ions present at the active site, two are brought in by the enzymatically active substrate ($Mg_2PP_i$) (Maeshima, 2000; Malinen et al, 2008). A water molecule is positioned for a nucleophilic attack on $PP_i$ and interacts with either one ($D^{16.39}$, resting state) or two aspartates ($D^{16.39}$ and $D^{6.43}$, active state) (Fig. 1B).

### Coupling funnel

The coupling funnel links the active site to the ion gate in the centre of the membrane (Fig. 1A) and couples $PP_i$ hydrolysis to the transport of $Na^+$, $H^+$ or both across the membrane (Kellosalo et al, 2012; Lin et al, 2012). A set of highly conserved charged residues form an ion pathway through the cytoplasmic half of the membrane, allowing ion translocation (Kellosalo et al, 2012; Lin et al, 2012).

### Ion gate and exit channel

The ion gate functions as an ion selectivity filter for pumping in the membrane-spanning protein region. Four residues, $E^{6.53/57}$, $D^{6.50}$, $S^{6.54}$, $D/N^{16.46}$, form a $Na^+/H^+$ binding site (Fig. 1C,D). Binding of $Mg_2PP_i$ to the active site requires a downward shift of helix 12 and corkscrew motion at helix 6 and 16, which affects the ion gate configuration of $K^+$-dependent $Na^+$-PPases (*Tm*PPase) and $K^+$-dependent $H^+$-PPases (*Vr*PPase) differently. In $K^+$-dependent $Na^+$-PPases, $K^{16.50}$ rotates out of the $Na^+$-binding site, making it available for ion binding (Fig. 1C). In $K^+$-dependent $H^+$-PPases, $K^{16.50}$ reorientation unmasks a proton binding site instead by leaving the side chain of $E^{6.57}$ stabilised only by a hydrogen bond to $S^{5.43}$ in a hydrophobic protein environment (Fig. 1D). It has been proposed that $E^{6.57}$ is protonated, thus linking structural differences at the ion gate between $K^+$-dependent $Na^+$-PPases and $K^+$-dependent $H^+$-PPases to the observed ion-pumping selectivity (Li et al, 2016). The *exit channel* below the ion gate has low sequence conservation but its properties are important in facilitating ion release into the extracellular space (Tsai et al, 2019).

### Dimer interface

The interface between monomers is formed by residues of the outer ring helices 10, 13 and inner ring helix 15 (Fig. 1A) that interact

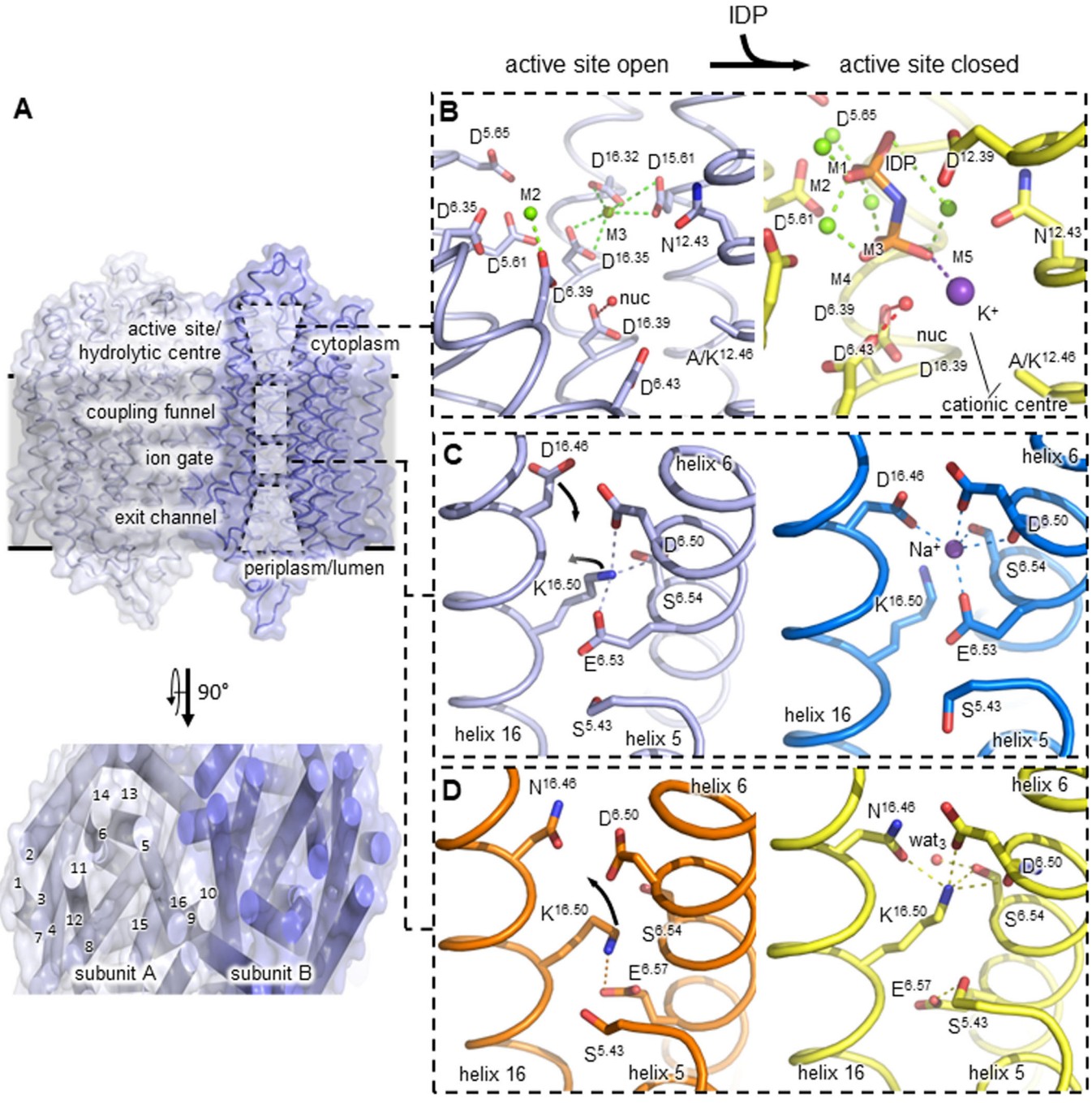

**Figure 1. Structural features of M-PPases.**

Protein colouring follows previous publications (Li et al, 2016; Vidilaseris et al, 2019) that used shades of yellow/orange for *Vr*PPase and shades of blue for *Tm*PPase structures. (**A**) Homodimeric M-PPase viewed from the membrane plane (top) with the functional core (active site, coupling funnel, ion gate, exit channel) highlighted by dashed boxes. Concentric ring arrangements of transmembrane helices (bottom) viewed from the cytoplasmic site. Loops were removed for clarity. (**B**) Close-up view of the *Tm*PPase (resting state: left panel (PDB: 4AV3) or *Vr*PPase (active state: right panel (PDB: 4A01)) active site with helix 11 removed for clarity. M1-5 is $Mg^{2+}$ (active state), M3 is inhibitory $Ca^{2+}$ (resting state). $K^+$ (purple sphere) is part of the cationic centre in $K^+$-dependent M-PPases (with $A^{12.46}$). The non-hydrolysable substrate analogue IDP (imidodiphosphate) is shown in orange. (**C, D**) Close-up view of the ion gate in $K^+$-dependent $Na^+$-PPases (resting state: left panel (PDB: 4AV3), active state: right panel (PDB: 5LZQ)) (**C**) or $K^+$-dependent $H^+$-PPases (product-bound: left panel (PDB: 5GPJ), active state: right panel (PDB: 4A01)) (**D**). $Na^+$ shown as purple sphere, structural water ($wat_3$) displayed as red sphere. Dashed lines highlight key interactions.

with the opposing subunit via hydrogen bonds and hydrophobic interactions (Kellosalo et al, 2012; Lin et al, 2012). The dimer interface has not typically been considered as key to the function of M-PPases as all of the catalytic machinery seems to be located in a single subunit (Kellosalo et al, 2012; Lin et al, 2012), but a growing body of structural and functional evidence points to (1) that M-PPases are functionally asymmetric, and (2) that the dimer interface mediates key inter-subunit interactions (Anashkin et al, 2021; Artukka et al, 2018; Vidilaseris et al, 2019) through coupled helix motions during the catalytic cycle (Li et al, 2016).

## Energy coupling

The chronological order of $PP_i$ hydrolysis and ion-pumping is a point of active discussion (Baykov, 2020; Li et al, 2016). The two opposing mechanisms of energy coupling either postulate pumping-before-hydrolysis or pumping-after-hydrolysis. The pumping-after-hydrolysis model, also called Mitchell-direct, postulates that the $H^+$ released from the nucleophilic water during $PP_i$ hydrolysis is the one pumped after $n$ cycles, where $n$ is the number of downstream ion-binding sites (Baykov, 2020). This was extended by a billiard-type mechanism to explain $Na^+$-transport in which the generated $H^+$ pushes $Na^+$ into the exit channel for pumping (Baykov et al, 2013). In contrast, the pumping-before-hydrolysis model, also called binding-change, favours a mechanism in which ion-pumping precedes hydrolysis and is initiated by binding of the substrate, triggering the closure of the active site and associated helical rearrangements. The transported ion may originate from the medium or preceding hydrolysis events and can explain both $H^+$- and $Na^+$-pumping (Li et al, 2016). The overall negative charge at the ion gate that results from pumping-before-hydrolysis would then promote the abstraction of a $H^+$ from nucleophilic water at the active site and thereby drive the hydrolysis of $PP_i$. The generated $H^+$ could enter the ion pathway and reset the ion gate (Li et al, 2016).

## The evolution of $K^+$-independence and ion-pumping selectivity

Two coupled changes are correlated with the evolution of $K^+$-independent $H^+$-PPases: $A^{12.46}K$ and $G/A^{12.49}T$ (Belogurov and Lahti, 2002). Of them, the change at position 12.46 is the one that defines $K^+$-dependence (Artukka et al, 2018): the $\varepsilon$-$NH_3^+$ of $K^{12.46}$ has been postulated to replace $K^+$ in the cationic centre both functionally and structurally (Fig. 1B). However, there has been no structural data available to support this idea. Moreover, although the $A^{12.46}K$ and $G/A^{12.49}T$ changes are tightly coupled evolutionarily, there is no functional role so far ascribed to the residue at position 12.49 (Belogurov and Lahti, 2002). It appears to be involved in $K^+$-binding as $A^{12.49}T$ single variants of $K^+$-dependent $H^+$-PPases show a threefold reduced affinity for $K^+$, but it remains unclear how changes at this position affect the cationic centre, which is ~10 Å away (Belogurov and Lahti, 2002). Alternatively, $G/A/T^{12.49}$ may play a crucial role in substrate inhibition as this regulatory mechanism is lost in M-PPases when interfering with the native state of the cationic centre, e.g., in $A^{12.49}K$ single variants of $K^+$-dependent $H^+$-PPases (Artukka et al, 2018).

In contrast to $K^+$-dependence, there are no conserved residue patterns that correlate with ion-pumping selectivity across *all*

M-PPase subclasses, but the C-terminal shift of a key glutamate at the ion gate of $K^+$-*dependent* M-PPases by one helix turn ($E^{6.53 \rightarrow 57}$) is coupled to a change in selectivity ($Na^+ \rightarrow H^+$) (Lin et al, 2012; Li et al, 2016). When the semi-conserved glutamate is located one helix turn down, $K^{16.50}$ continues to block the $Na^+$-binding site upon substrate binding at the active site, while $E^{6.57}$ reorientates and can now accommodate a proton (Fig. 1D). However, this model fails to explain ion-pumping selectivity in $K^+$-independent $H^+$-PPases or $K^+$-dependent $Na^+,H^+$-PPases, as both contain $E^{6.53}$ (Table 1). It might make more sense to consider $E^{6.53}$ the conserved position, with mutations in $K^+$-dependent $H^+$-PPases containing $E^{6.57}$.

Taken together, there are many unanswered questions about mechanistic key functions: what is the mechanism of energy coupling and of ion-pumping selectivity?; what is the structure of $K^+$-independent M-PPases?; and what is the structural/functional basis of catalytic asymmetry? To address them, we solved the first structure of a $K^+$-independent M-PPase (from the thermophile *Pyrobaculum aerophilum*, *Pa*PPase), performed enzymatic assays on native *Pa*PPase and three variants ($A^{12.46}K$ and $A^{12.49}T$, and the double mutant), and conducted both electrometric and time-resolved crystallographic studies on *Tm*PPase, a $K^+$-dependent $Na^+$-PPase.

# Results

## Structure of *Pa*PPase

Purified wild-type *Pa*PPase (Appendix Fig. S1A,B) readily crystallised in vapour diffusion set-ups, but despite extensive optimisation efforts, the diffraction was anisotropic. The data were thus submitted to the STARANISO webserver (Tickle et al, 2018), and the structure was solved by molecular replacement (MR) using a modified *Tm*PPase:$Mg_5$IDP (PDB: 5LZQ) search model (IDP, imidodiphosphate) with all loops and heteroatoms removed. This yielded a structure with one *Pa*PPase homodimer molecule per asymmetric unit and resolutions of 5.3 Å, 4.1 Å and 3.8 Å along h, k and l, respectively (Table EV1). After initial refinement, positive $mF_o$-$dF_c$ density was observed at 3 σ in both subunits of the *Pa*PPase active site, at the ion gate (Appendix Fig. S2A) and in the dimer interface. We built a $Mg_5$IDP complex into the active site, as seen in other IDP-bound M-PPase structures, and two water molecules in regions with excess positive $mF_o$-$dF_c$ density that bridge between $loop_{5-6}$ and the metal cage (Appendix Fig. S2B). A structural water was built into the positive $mF_o$-$dF_c$ density at the ion gate as in the high-resolution *Vr*PPase structure and a sulfate molecule ($SO_4^{2-}$) was placed at the dimer interface (Appendix Fig. S2B). The electron density maps improved throughout refinement and the final $R_{work}$/$R_{free}$ was 28.9%/31.1% with appropriate stereochemistry for this resolution range (3.8–5.3 Å).

## Structural overview and comparison of M-PPase structures

The *Pa*PPase structure is in the $Mg_5$IDP-bound state with $loop_{5-6}$ closed and a structural water located at the ion gate (Fig. 2). In what follows, structural alignments and root mean square deviation (r.m.s.d.) calculations are based on the Cα atom of subunit A (both

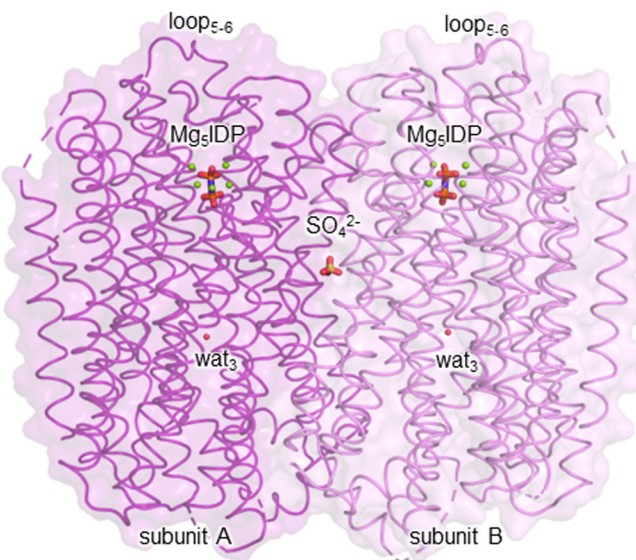

**Figure 2. Overview of the *Pa*PPase:Mg$_5$IDP structure.**

Subunits, loop$_{5-6}$ and ligands/structural water molecules are annotated.

subunits are nearly identical; r.m.s.d./C$_\alpha$: 0.27 Å), unless stated otherwise. The overall structure is very similar to other published M-PPase structures with an average r.m.s.d ($\overline{\text{r.m.s.d./C}\alpha}$) of 1.37 ± 0.18 Å to IDP-bound structures, 1.41 ± 0.15 Å to product-bound structures, and a r.m.s.d./C$_\alpha$ of 1.67 Å to the resting-state structure (Appendix Table S2). In general, outer ring helices display more variability ($\overline{\text{r.m.s.d./C}\alpha_{OR}}$: 2.01 ± 0.67 Å) than inner-ring helices ($\overline{\text{r.m.s.d./C}\alpha_{IR}}$: 1.27 ± 0.32 Å) when compared to *Tm/Vr*PPase:Mg$_5$IDP.

Alignment-independent inter-atom difference distance matrices (DiDiMa) (Grant et al, 2006) highlighted outer ring helices 13–14 (and 2–3 in *Vr*PPase but not *Tm*PPase) as regions with major structural differences when comparing identical enzyme states (Fig. EV1A). In published IDP-bound structures, helices 13–14 are consistently bent halfway through the membrane by about 9° to remain near the cytoplasmic regions of helix 5 (Fig. EV1B). This enables propagation of motions from the inner to the outer ring and into the other subunit (indicated by an apostrophe) via E$^{5.71}$–R$^{13.62}$–R/I/K'$^{10.33}$ (the interaction to R/I/K$^{10.33'}$ is *via* the backbone carbonyl, explaining the lack of sequence conservation; Appendix Fig. S4A,B) and was linked to loop$_{5-6}$ closure and subsequent helical rearrangements (Li et al, 2016). In our new structure, the E$^{5.71}$-R$^{13.62}$-R/I/K'$^{10.33}$ interaction is lost (Appendix Fig. S4C): the cytoplasmic regions of helices 13–14 are straightened (Fig. EV1B), resembling resting state (Fig. EV1C) and product-bound (Fig. EV1D,E) structures despite having IDP bound. This suggests a different role for helix 13–14 bending in M-PPase function than previously thought (see "Discussion").

The only inner ring (IR) helix with above-average conformational differences to previously published IDP-bound structures is helix 5 ($\overline{\text{r.m.s.d./C}\alpha_{h5}}$: 1.70 ± 0.13 Å versus $\overline{\text{r.m.s.d./C}\alpha_{IR}}$: 1.27 ± 0.32 Å), around which outer ring helices 2–3 and 13–14 cluster. Helix 5 is straighter than in other M-PPases (Fig. 3A; Appendix Table S3), which also straightens helices 13–14 (cytoplasmic side, Fig. 3B) and 2–3 (periplasmic side, Fig. 3C) by

pushing them away from the inner ring. In addition, helix 5 is more tightly wound in *Pa*PPase:Mg$_5$IDP due to the presence of twice as many 3$_{10}$ hydrogen bonds around S$^{5.43}$ and towards its flanking periplasmic segment (Appendix Table S4). Consequently, the side-chain orientations are different in this region compared with *Vr/ Tm*PPase:Mg$_5$IDP. This is particularly interesting, as S$^{5.43}$ is part of the enzymatic core region defining ion selectivity (see "Discussion").

## Structural and functional characterisation of K⁺ independence in *Pa*PPase

The coordination of the Mg$_5$IDP complex (Fig. 4A) at the active site by acidic residues is almost identical to *Tm/Vr*PPase:Mg$_5$IDP (r.m.s.d./C$_\alpha$: 0.81/0.73 Å, alignment of catalytic residues in the active site of subunit A), and the hydrolytic pocket volume is about 1200 Å$^3$ for all three structures. However, there are some interesting structural changes in this protein region as compared to previously solved K⁺-dependent M-PPases with A$^{12.46}$ and A/G$^{12.49}$:

We noticed that T$^{12.49}$ interacts with the conserved residue D$^{11.50}$ in *Pa*PPase (Fig. 4B), a direct result of the coupled A/G$^{12.49}$T change in K⁺-independent M-PPases. This appears to lead to a cascade of structural changes at nearby residues (D$^{11.50}$–M$^{6.47}$) to avoid clashing (Fig. 4C); which ultimately affects the orientation of D$^{6.50}$ and the helical geometry at D$^{6.43}$, both catalytical essential general bases of enzyme core regions (see "Introduction").

We then measured the activity of wild-type, K$^{12.46}$A, T$^{12.49}$A and double variants. As expected, hydrolysis by wild-type *Pa*PPase is not activated by K⁺ (Fig. 4D) but is inhibited by substrate (Fig. 4E), with very weak binding of the second PP$_i$ (Table 2). The nature of the kinetic scheme means that the second binding site reported for PP$_i$ must be in the other monomer, as it can be hydrolysed (it corresponds to the E$_2$S$_2$ complex). Half-of-the-sites reactivity has been reported before on a variety of M-PPase (Artukka et al, 2018; Vidilaseris et al, 2019; Anashkin et al, 2021). The K$^{12.46}$A variant is weakly activated by K⁺ (Fig. 4D). Without K⁺, the K$^{12.46}$A variant displays Michaelis–Menten kinetics; in the presence of 100 mM K⁺, it has similar substrate inhibition as wild-type (Fig. 4E) except that V$_2$ is now zero (Table 2). Surprisingly, the T$^{12.49}$A variant is essentially inactive (Fig. 4D), while the double mutant no longer shows signs of substrate inhibition (Fig. 4E): conventional Michaelis–Menten kinetics provide acceptable fits to the data (Table 2). Taken together, these results suggest that helix 12, the site of the largest motion in the active site during catalysis with key residues K$^{12.46}$ and T$^{12.49}$ (Li et al, 2016), may play a crucial role in inter-subunit communication and, furthermore, that the observed half-of-the-sites reactivity (Vidilaseris et al, 2019; Artukka et al, 2018; Anashkin et al, 2021) may be key to understanding the true catalytic cycle (see "Discussion").

We were not able to confirm definitively the kinetic data that place K$^{12.46}$ at the heart of K⁺-independence structurally, as residues with flexible side chains such as lysines lacked sufficient electron density at the given resolution (Appendix Fig. S3). There is an ongoing debate among crystallographers about how to handle disordered sidechains in model building (Lamb et al, 2015). One can simply remove side chains for which there is no experimental evidence, i.e., electron density, to avoid bias and overinterpretation of data; however, the truncation of side chains may lead to confusion about amino acid identity for non-expert viewers, does

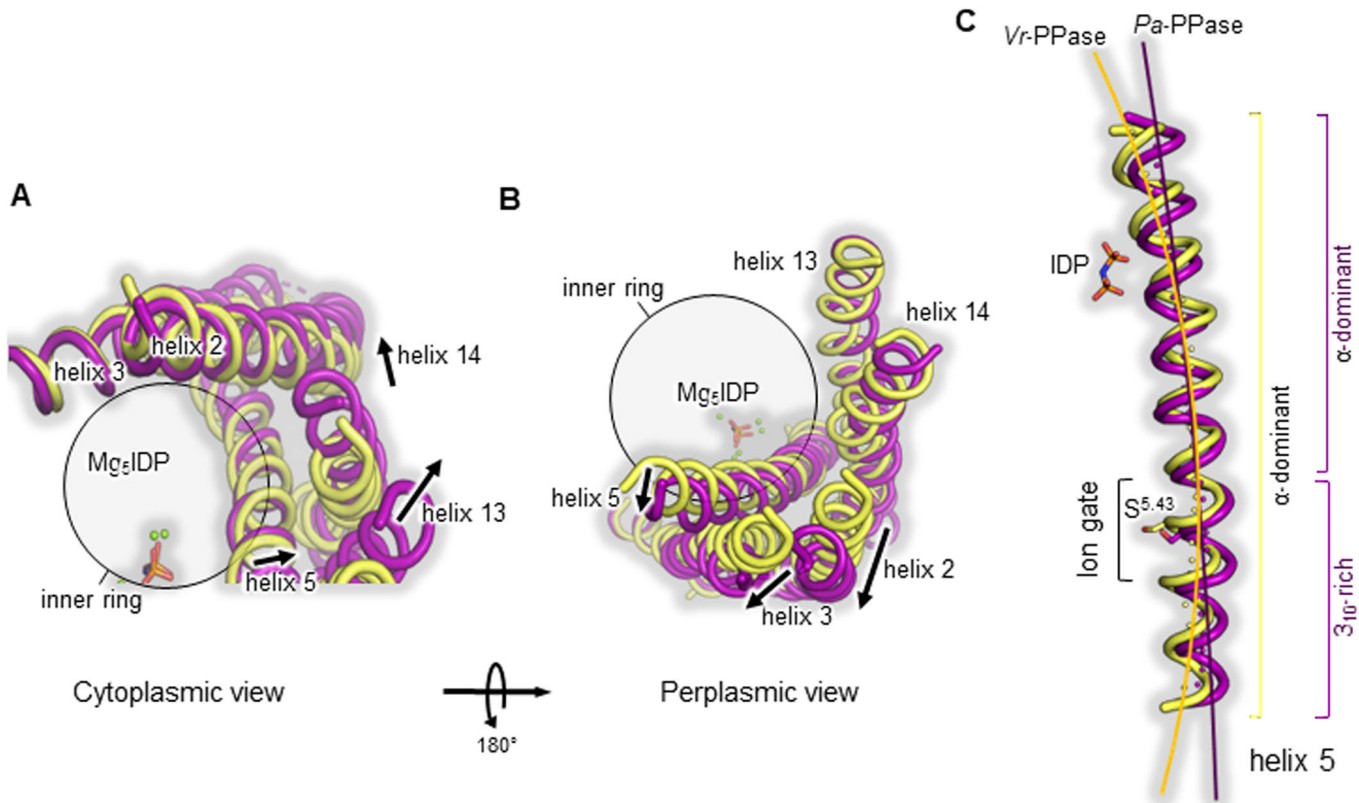

**Figure 3. Comparison of helices 2–3, 5 and 13–14 orientations in *Pa*PPase:Mg₅IDP to *Vr*PPase:Mg₅IDP.**

*Pa*PPase:Mg₅IDP (this study) is shown in purple and *Vr*PPase:Mg₅IDP (PDB: 4A01) is shown in yellow. Major conformational changes are indicated by arrows. The alignment is based on Cα atoms of subunit A of the respective structures. (**A**) Straightening of helix 5 pushes helices 13–14 away from the inner ring on the cytoplasmic site. (**B**) Straightening of helix 5 pushes helix 2–3 away from the inner ring on the periplasmic site. (**C**) Close-up view of helix 5, showing its straightening in *Pa*PPase:Mg₅IDP compared to *Vr*PPase:Mg₅IDP. Helix straightening is highlighted by a curve running through the centre of helix 5, which was manually fitted to the local helix origin points that were obtained from HELANAL-Plus analysis and are displayed as spheres in the helix centre (Kumar and Bansal, 2012).

not reflect the real chemical entity (Lamb et al, 2015) and may lead to other errors in interpretation—the encroachment of other residues into the region with missing atoms. It is reasonable to argue that the sidechains should be included in a refined structure as they are most definitely chemically present and that the resolved atoms both locally and globally constrain the orientation of individual side-chain atoms. Following this rationale, the side chain is placed in the stereochemically most plausible pose, which gives a good approximation of at least one potential real position, especially if residues adjacent in three dimensions are well-defined. This approach avoids confusion about amino acid identity, but obliges the viewer to check electron density maps and B-factors, which will refine to high values in the absence of electron density, to determine the model reliability in the region of interest (Lamb et al, 2015). We therefore decided to model poorly defined side chains in catalytic key regions in their stereochemically most plausible pose (Appendix Tables S5 and 6) and allow their refinement to high B-factors, as they are chemically present in the enzyme, but explicitly flag the absence of sufficient density in the text where appropriate and do not base our mechanistic models on such side-chain orientations. For K$^{12.46}$, this indeed places ε-NH$_3^+$ of K$^{12.46}$ at the K$^+$-binding site (Fig. 4B), which would explain K$^+$-independence. It is worth noting that the most likely poses are

clearly energetically separated even from the second most likely poses in terms of clash score and hydrogen bonds (Appendix Table S5). Nevertheless, a more robust analysis requires higher-resolution data.

## Ion selectivity in K$^+$-independent H$^+$-PPases

The structure of the ion gate must hold the explanation to ion selectivity, but the current model, that the position of the semi-conserved glutamate alone defines ion selectivity does not hold for K$^+$-independent H$^+$-PPases (see "Introduction"). Our *Pa*PPase:Mg₅IDP structure provides the first data on the ion gate set-up of K$^+$-independent H$^+$-PPases with E$^{6.53}$ and shows that the 3D orientation of ion gate residues D$^{6.50}$, S$^{6.54}$, N$^{16.46}$ and E$^{6.53}$ in particular *all* resemble the structure of *Vr*PPase:Mg₅IDP, a H$^+$-pump with E$^{6.57}$, and not *Tm*PPase:Mg₅IDP, a Na$^+$-pump with E$^{6.53}$, despite the shift of the semi-conserved glutamate (Fig. 5A). This raises the question: What causes E$^{6.53}$ to reorient in *Pa*PPase so that it forms the same interactions as E$^{6.57}$ in *Vr*PPase, not as E$^{6.53}$ in *Tm*PPase (Fig. 5B)?

In *Pa*PPase:Mg₅IDP, helix 5 is straightened (see above) and moved out of the protein core at the ion gate by about 2 Å compared with other IDP-bound M-PPases (Fig. 5C,D). This allows S$^{5.43}$ to hydrogen bond to E$^{6.53}$, as occurs with E$^{6.57}$ in

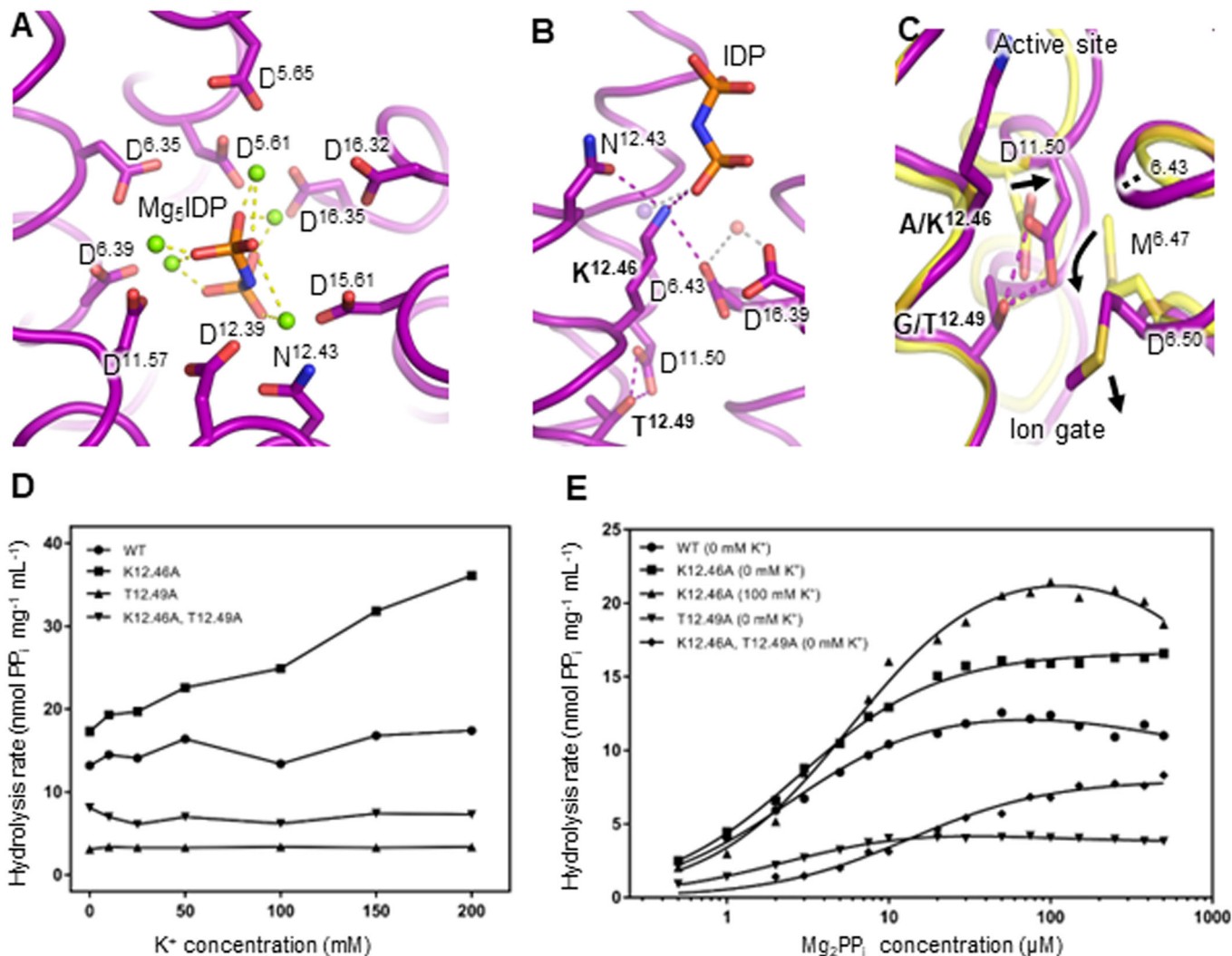

**Figure 4. Structural overview and functional characterisation of the cationic centre in the *Pa*PPase active site.**

(A) Active site with IDP coordinated (dashed lines) in a $Mg^{2+}$ metal cage (green spheres). (B) $K^+/K^{12.46}$ cationic centre with $K^+$ (transparent purple sphere) and nucleophilic water (transparent red sphere) modelled into the structure based on its position in $Vr$PPase:$Mg_5$IDP (PDB: 4A01). $K^{12.46}$ is not defined in the electron density but modelled in the stereochemically most plausible pose (Appendix Table S5), which would allow it to substitute for $K^+$. (C) Comparison of the cationic centre of PaPPase:$Mg_5$IDP (purple, this study) and VrPPase:$Mg_5$IDP (yellow, PDB: 4A01). Major structural changes caused by the $G^{12.49}$T change in $K^+$-independent M-PPases are highlighted by black arrows. Dotted line indicates change in helical geometry at $D^{6.43}$. Key residues of $K^+$-independence are labelled in bold. Their interaction is shown by dashed lines. (D) Potassium dependency of $PP_i$ hydrolysis of wild-type and variant $Pa$PPase. (E) Wild-type and variant $Pa$PPase kinetics (hydrolysis rate versus logarithmic $Mg_2PP_i$ concentration) in the absence and presence of 100 mM $K^+$. All data were collected in the presence of 5 mM free $Mg^{2+}$. Wild-type (0 mM $K^+$), $K^{12.46}$A (100 mM $K^+$), and $T^{12.49}$A (0 mM $K^+$) show the best fit to Eq. (1), while $K^{12.46}$A (0 mM $K^+$) and $K^{12.46}$A,$T^{12.49}$A (0 mM $K^+$) show the best fit to the Michaelis–Menten equation. Source data are available online for this figure.

$Vr$PPase:$Mg_5$IDP (Fig. 1D). In $Tm$PPase, helix 5 is closer to helix 6 and helix 16, forcing position 6.53 to point away from $S^{5.43}$, thereby contributing to the formation of a $Na^+$-binding site ($Tm$PPase:M-$g_5$IDP) (Fig. 5C).

### An ion-binding site at the dimer interface

The dimer interface of $Pa$PPase is formed by helices 10, 13 and 15 and somewhat different to other M-PPases (Fig. EV2A). Usually, non-polar amino acids are conserved at position 10.44 (97.6% conserved) and 15.49 (95.2% conserved). In $Pa$PPase these are

tyrosine and arginine, respectively (Fig. EV2B). The additional hydrogen-bonding potential and positive charge leads to the formation of an anion-binding site. We modelled $SO_4^{2-}$ from the crystallisation solution to mediate the inter-subunit communication of $Y^{10.44}$, $Y^{13.44}$, and $R^{15.40}$ in $Pa$PPase:$Mg_5$IDP, but this may be $P_i$ (not present in crystallisation) under physiological conditions.

### Direct observation of asymmetric $PP_i$ binding in $Tm$PPase

Structural information about inter-subunit communication and functional asymmetry is essential to resolve unanswered questions

**Table 2. Kinetic parameters for PP$_i$ hydrolysis of PaPPase.**

| Parameter | Value | | | | |
|---|---|---|---|---|---|
| | Wild-type (0 mM K$^+$) | K$^{12.46}$A (0 mM K$^+$) | K$^{12.46}$A (100 mM K$^+$) | T$^{12.49}$A (0 mM K$^+$) | K$^{12.46}$A, T$^{12.49}$A (0 mM K$^+$) |
| Equation | Substrate inhibition | Michaelis–Menten | Substrate inhibition | Substrate inhibition | Michaelis–Menten |
| $V_{max}$ (nmol PP$_i$ mg$^{-1}$ min$^{-1}$) | | 16.64 ± 0.12 | | | 7.98 ± 0.21 |
| $V_1$ (nmol PP$_i$ mg$^{-1}$ min$^{-1}$) | 12.89 ± 0.34 | | 23.34 ± 0.60 | 4.739 ± 0.12 | |
| $V_2$ (nmol PP$_i$ mg$^{-1}$ min$^{-1}$) | 9.44 ± 3.22 | | 0 | 3.78 ± 0.094 | |
| $K_m$ (µM) | | 2.81 ± 0.08 | | | 13.48 ± 1.12 |
| $K_{m1}$ (µM) | 2.42 ± 0.21 | | 5.86 ± 0.39 | 2.30 ± 0.15 | |
| $K_{m2}$ (µM) | 449.8 ± 828.4 | | 2223 ± 578.4 | 43.99 ± 23.14 | |

about variable ion-pumping selectivity and, potentially, energy coupling in M-PPases (see "Introduction"). Unfortunately, structural data on inhibited, i.e., symmetric, enzyme can only provide limited insight. We therefore decided to study the K$^+$-dependent Na$^+$-PPase from *T. maritima* (*Tm*PPase) using a time-resolved cryo-trapping approach to be able to map physiologically more relevant enzyme states.

The catalytic turnover ($k_{cat}$) of purified *Tm*PPase that was crystallised (Appendix Fig. S5A) in conditions suitable for time-resolved studies (i.e., no inhibitors) was 282-fold lower ($k_{cat}$: 0.16 ± 0.05 s$^{-1}$ at 20 °C) compared to ideal reaction conditions ($k_{cat}$: 45.13 ± 3.59 s$^{-1}$ at 71 °C) (Appendix Fig. S5B–D and Appendix Table S7), of which about a factor of 32–234 may be ascribed to the change in temperature assuming a Q$_{10}$ of 2–3 as for most biological systems (Blehrádek, 1926). The Na$^+$-PPase was thus active and, importantly, the substrate turnover sufficiently slow for a manual, single-crystal time-resolved cryo-trapping approach in which the reaction was initiated by soaking-in activating Na$^+$ (see "Methods").

The collected time-resolved *Tm*PPase datasets ($t = 0$–3600 s) were anisotropic (Table EV2) and therefore run through the STARANISO webserver for anisotropy correction. Non-soaked reference crystals ($t = 0$ s) yielded a structure with a resolution of 2.65 Å along h, 3.32 Å along k and 3.79 Å along l at best (Table EV2). To avoid bias, the *Tm*PPase structures were solved by molecular replacement using the *Tm*PPase:CaMg resting-state structure (PDB: 4AV3) as a search model with heteroatoms removed. The 0-s structure has one homodimer molecule per asymmetric unit, refined to an R$_{work}$/R$_{free}$ of 23.8/27.4% and was very similar to the inhibited *Tm*PPase:CaMg structure (r.m.s.d./C$_\alpha$: 0.41 Å). To check for asymmetry, subunits A and B were refined individually, but remained identical at this resolution, with an r.m.s.d./C$_\alpha$ of 0.21 Å. Despite the presence of PP$_i$ in the crystallisation condition, it was not located at the active site, nor did it bind anywhere else.

The diffraction quality of crystals that were soaked in Na$^+$-trigger solution for up to 60 s was similar to non-soaked reference crystals. Interestingly, the diffraction quality declined abruptly in datasets collected at >60 s (Fig. 6A). The diffraction limits for the 300-, 600- and 3600-s time points were 3.77 Å, 3.84 Å and 4.53 Å in the best direction, respectively. The changes in diffraction quality also aligned with BLEND analysis (Aller et al, 2016) of the linear cell variability (Fig. 6B) in which the 0–60-s datasets cluster well (with linear cell variabilities (LCV) of <2.8%). Following inspection of enzyme key regions and structural alignments, it

became clear that all of the data from 0 to 60 s could be combined into one, yielding a grouped structure of *Tm*PPase in the resting state that is essentially identical to the individual 0-s (r.m.s.d./C$_\alpha$: 0.33 Å) and *Tm*PPase:CaMg (r.m.s.d./C$_\alpha$: 0.29 Å) structure. This led to improved data quality parameters, including resolution and completeness, while R$_{pim}$ remained stable and within the generally accepted limit of ~5% (Table EV2). The structure was solved at 2.54 Å along h, 2.95 Å along k and 3.38 Å along l with improved B-factors (108.65–70.28 Å$^2$) and R$_{work}$/R$_{free}$ values (21.95/23.61%) as compared to the 0-s structure (23.81/27.40%). The improvement in data quality parameters also translated into better electron density maps, so we could model most side chains and build additional key loop regions, for example, loop$_{5-6}$. Upon calculating difference Fourier maps for the later time points using the 0–60 s structure for phases, we were able to observe, for the first time, significant asymmetry in an active M-PPase, corresponding to the first steps in the catalytic cycle. We then combined all datasets obtained at the same time point, which likewise improved electron density maps and data quality parameters (Table EV2). While it became increasingly difficult to model side-chain orientations at >60 s, we were able to reliably model main chain density. Most importantly, the data clearly show that the M-PPase active site is asymmetrically occupied in the 300 and 600 s structure, i.e., there is strong positive mF$_o$-dF$_c$ density at 3 σ in subunit A but none in subunit B, and the position of that positive density changes between these two time points (Figs. 6C,D and 7A).

At 300 s, only full-occupancy PP$_i$ (and 4 Mg$^{2+}$ ions of the metal cage) provided an acceptable fit to the active site density (Fig. 7B) at the given resolution. All other buffer or crystallisation molecules of appropriate size including lower-occupancy PP$_i$ or (P$_i$)$_2$ introduced strong positive and/or negative mF$_o$-dF$_c$ density as well as a higher R$_{work/free}$ after refinement. At this time point, all helix orientations remained highly symmetrical between both subunits (r.m.s.d./C$_\alpha$ subunit A vs. B: 0.23 Å), and very similar to the resting-state *Tm*PPase:CaMg (r.m.s.d./C$_\alpha$: 0.35 Å) and the 0–60-s structures (r.m.s.d./C$_\alpha$: 0.13 Å) despite having PP$_i$ bound. After 600 s, the position of the positive density at the active site of subunit A changed but was still fit best by PP$_i$ (Fig. 7B) when modelled in the canonical position for hydrolysis based on the *Tm*PPase:Mg$_5$IDP reference structure (Fig. EV3A). Consequently, subunit A transitioned into the active conformation, i.e., downward shift of helix 12 and reorientation of helix 6 and 16, as seen in *Tm*PPase:Mg$_5$IDP (PBD: 5LZQ), *Pa*PPase:Mg$_5$IDP (this study) and *Vr*PPase:Mg$_5$IDP (PDB: 4A01) structures, all while subunit B remained in the resting state (r.m.s.d./C$_\alpha$ subunit A vs. B: 0.78 Å).

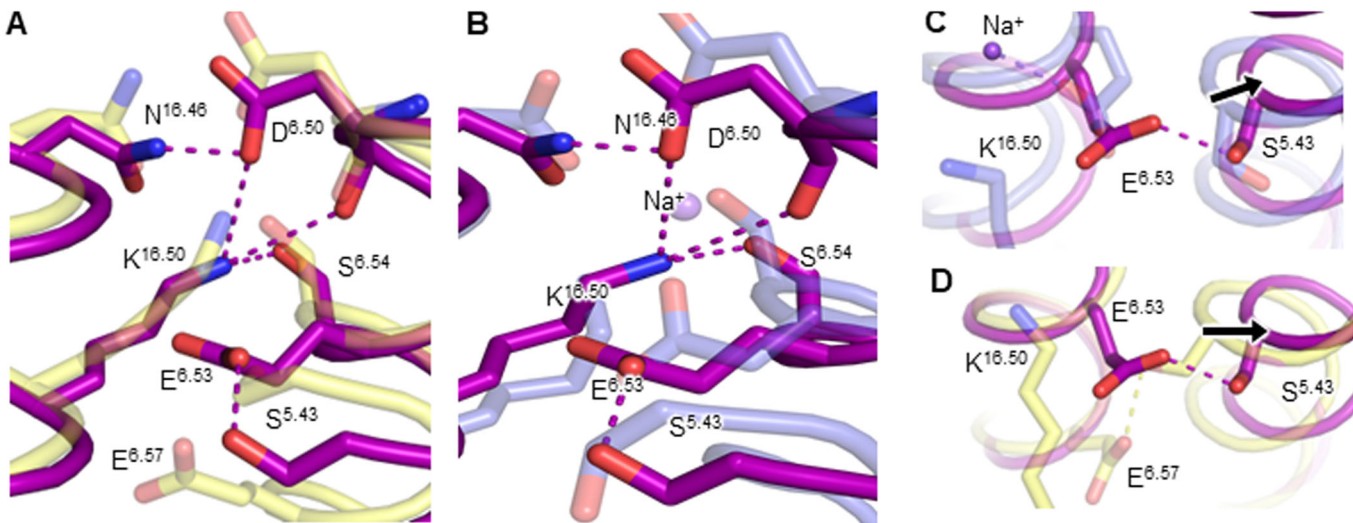

**Figure 5. Structural overview of the ion gate in *Pa*PPase:Mg$_5$IDP.**

(**A**) Comparison of the *Pa*PPase:Mg$_5$IDP (purple, this study) and *Vr*PPase:Mg$_5$IDP (yellow, PDB: 4A01) ion gate structures. (**B**) Comparison of the *Pa*PPase:Mg$_5$IDP and *Tm*PPase:Mg$_5$IDP (blue, PDB: 5LZQ) ion gate structures. (**C**) Close-up view and comparison of the semi-conserved glutamate (E$^{6.53/57}$) orientation and helix 5 conformation in *Pa*PPase:Mg$_5$IDP and *Tm*PPase:Mg$_5$IDP. (**D**) Close-up view and comparison of the semi-conserved glutamate (E$^{6.53/57}$) orientation and helix 5 conformation in *Pa*PPase:Mg$_5$IDP and *Vr*PPase:Mg$_5$IDP. Dashed lines show the coordination of key residues involved in ion selectivity. K$^{16.50}$ is not defined in the electron density but modelled in the stereochemically most plausible pose (Appendix Table S6), which aligns with its orientation in *Vr*PPase:Mg$_5$IDP. Major structural changes are indicated by black arrows.

In the 3600-s structure, both active sites appear to be occupied (Figs. 6C,D and 7A). The loss of resolution upon binding makes analysis very difficult indeed, but the mF$_o$-dF$_c$ maps are not inconsistent with the idea that, at this point, one subunit binds PP$_i$ and the other (P$_i$)$_2$ (Fig. 7B).

We are aware that higher-resolution data would enable a more in-depth analysis of the mechanistic details of pumping, excess substrate inhibition and inter-subunit communication. Extensive efforts were dedicated to improving the resolution, and hundreds of crystals were tested. Only a handful of crystals were obtained that diffracted sufficiently to be combined and yield the time-resolved structures presented in this study. At the obtained resolution, the structures still give a reliable snapshot of the global enzyme conformation at different time points and a number of clear mechanistic details are observed. The 300- and 600-s structure clearly indicate asymmetric occupation of the active site for the first time, while the 3600-s data demonstrates unrestricted access to the active site to prove that that asymmetric substrate binding is not simply a crystallographic artefact.

These unambiguous observations are consistent with the half-of-the-sites reactivity seen in kinetic assays (Anashkin et al, 2021; Artukka et al, 2018; Vidilaseris et al, 2019); they provide snapshots of a structural binding pathway, and support a new comprehensive kinetic model of catalysis (see "Discussion"). There were no observable significant changes in other regions of the protein, but the semi-conserved glutamate E$^{6.53}$ appears to be flexible in all the structures as indicated by the absence of 2mF$_o$-dF$_c$ density, even in the high-resolution 0–60 s structure, or negative density when modelled as seen in *Tm*PPase:CaMg (Fig. EV3C). All other side-chain orientations at the ion gate were well-defined in the 2mF$_o$-dF$_c$ map up to the 300-s time point. Typically, the semi-conserved glutamate is well ordered, and side-

chain density is visible at much lower resolutions (see *Pa*PPase structure, Appendix Fig. S2B). Its flexibility in low Na$^+$ conditions, if real, has implications for the ion-pumping selectivity at sub-physiological Na$^+$ concentrations in K$^+$-dependent Na$^+$-PPases (see "Discussion").

## Energy coupling

We further used electrometric measurements to study energy coupling of PP$_i$ hydrolysis and Na$^+$ pumping in *Tm*PPase. In this experiment, currents are generated when ions cross the membrane, so the measured current is the sum of the currents from all active proteins on the sensor. In the presence of 200 μM K$_2$HPO$_4$ as a negative control, the current was about 0.025 nA (Fig. 8A): phosphate can bind in the active site as shown in the structure of *Tm*PPase:Mg$_4$P$_{i2}$ (PDB: 4AV6) and *Vr*PPase:Mg$_2$P$_i$ (PDB: 5GPJ) but does not cause ion-pumping. A positive signal of 0.24 ±0.005 nA was detected after addition of 100 μM substrate (K$_4$PPi) and reached its maximum within ~0.1 s, i.e., in the dead time of the machine under the conditions used (Fig. 8). This is at least 12–25 times faster than the hydrolysis rates at 20 °C under similar lipidated conditions (Appendix Fig. S5C and Appendix Table S7). When the substrate was replaced by 100 μM of the non-hydrolysable analogue IDP, the signal was reduced by about half to 0.09 ± 0.009 nA. This finding has strong implications for determining the mechanisms of energy coupling in M-PPases as it favours a pumping-before-hydrolysis model (see "Discussion"). The signals decayed within 2.1–3.4 s, corresponding to *Tm*PPase entering a state that temporarily could no longer pump Na$^+$ at a sufficient rate to generate a signal. The current decay curves (Fig. 8B,C) were well fit by a single exponential with similar decay rates (*k*) for PP$_i$ (2.1 s$^{-1}$) and IDP (3.4 s$^{-1}$) (Table 3).

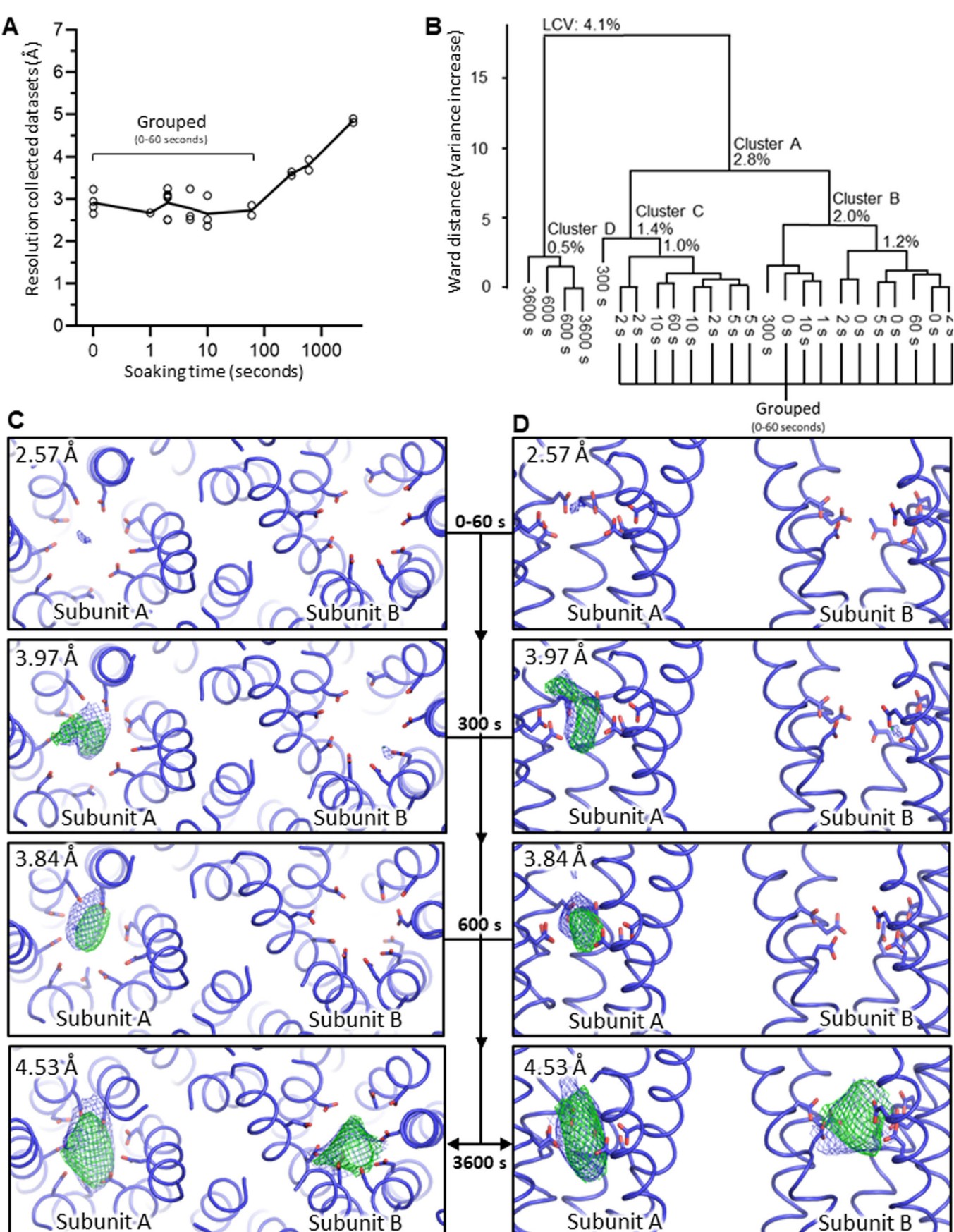

Figure 6.  Characterisation of time-resolved *Tm*PPase datasets.

(A) Diffraction quality at different time points. Each collected dataset is represented by a circle with the diffraction in the best direction plotted. The mean resolutions of each time point are connected by a black line. (B) Dendrogram of BLEND analysis to identify isomorphous time-resolved datasets. Nodes of the four biggest cluster are labelled with the linear cell variability (LCV). (C, D). *Tm*PPase active site from the cytoplasmic site (top down view) (C) or from the membrane plane (side view) (D) of combined datasets at different time points with $2mF_o\text{-}dF_c$ density (blue) and $mF_o\text{-}dF_c$ density (red/green) for ligand shown. The $2mF_o\text{-}dF_c$ density is contoured at 1 σ (0–60 s–600 s) or 2 σ (3600 s) and the $mF_o\text{-}dF_c$ density is contoured at 3 σ. The active site status changes from symmetrically empty, to asymmetrically occupied and finally "symmetrically" occupied. The resolution of the combined structures at each time point is stated. Source data are available online for this figure.

# Discussion

Our study has clearly indicated that M-PPases show anti-cooperative behaviour: productive substrate binding cannot happen in both sites at the same time (Table 2), consistent with earlier studies (Anashkin et al, 2021; Artukka et al, 2018; Vidilaseris et al, 2019). Artukka and co-workers proposed a model where binding in subunit A converts subunit B into a conformation that prevents substrate binding - even though all published structures of M-PPases with IDP show symmetrical binding to both subunits (Artukka et al, 2018). Vidilaseris et al, who identified an allosteric inhibitor of *Tm*PPase, suggest that motions of loops near the exit channel lead to asymmetry and play a role in intra-subunit communication (Vidilaseris et al, 2019). In their structure, these changes create a binding site for the allosteric inhibitor N-[(2-amino-6-  benzothiazolyl)   methyl]-1H-indole-2-carboxamide (ATC) in subunit A and prevent full motion in subunit B.

These are the key facts from which we endeavour to synthesise a comprehensive model of M-PPase catalysis. An ideal model would explain (a) half-of-the-sites reactivity; (b) energy coupling of hydrolysis and ion-pumping; (c) varying ion selectivity; and (d) how certain pumps can pump both $Na^+$ and $H^+$ using the same machinery. It is our contention that one, unified model that puts inter-subunit communication at the heart of the catalytic cycle explains all of these. Our evidence for this new model comes from kinetic data (Table 2) (Anashkin et al, 2021; Artukka et al, 2018; Vidilaseris et al, 2019), electrometric measurements (Fig. 8) and the time-resolved cryo-trapped structures (Fig. 6), which allowed us to map the global enzyme conformation and active site occupation throughout the reaction cycle for the first time.

## Half-of-the-sites reactivity

We start with a structural explanation of half-of-the-sites reactivity (functional asymmetry) in the context of our new data. *Pa*PPase, and $K^+$-independent M-PPases in general, have additional hydrogen bonding between helix 11 and 12 through $T^{12.49}\text{-}D^{11.50}$ (*Plasmodium spp.* $S^{11.50}$), that when lost in $T^{12.49}A$ mutants, abolishes activity. Moreover, the $K^{12.46}A$ mutation eliminates substrate inhibition and thus half-of-the-sites reactivity in *Pa*PPase that can only be restored by the addition of $K^+$. We propose that a positive charge in this position and the motion of helix 12 are key parts of communication. A second communication network is from helix 5 to helix 13 (Fig. 3B,C) and propagated through helix 10 (Appendix Fig. S4). This is in line with the model of intra-subunit ion gate to ion gate communication via exit channel loops, particularly $loop_{12\text{-}13}$, that was proposed by Vidilaseris et al (2019).

The asymmetric time-resolved *Tm*PPase structures (Figs. 6C and 7) also support half-of-the-sites reactivity and further indicate a highly ordered reaction mechanism. Ion binding is the first event,

as there is no evidence of $PP_i$ in any of the structures before 60 s post $Na^+$-addition, nor in the grouped 0–60-s structure (Fig. 6C), and the ion gate $E^{6.53}$ appears disordered (Fig. EV3B): $Na^+$ binding precedes substrate binding. There is also no evidence of any $PP_i$ binding in subunit B, which remains in the resting state up to 600 s after reaction initiation with $Na^+$, while subunit A has $PP_i$ asymmetrically bound. This changes in the 3600-s structure in which both active sites are occupied. In the context of a half-of-the-sites reactivity mechanism, we propose that this represents a configuration in which $PP_i$ is bound in one subunit and $(P_i)_2$ in the other (Fig. 9). However, the resolution of the structure is only 4.53 Å: higher-resolution time-resolved structural data is required to confirm this.

## Pumping-before-hydrolysis

We continue with the controversy discussed mechanism of energy coupling in M-PPase. *Tm*PPase catalysis as measured by phosphate production has a $k_{cat}$ of $0.2–0.4 \, s^{-1}$ at 20 °C (Appendix Table S7) but maximal signal in the electrometric measurements is reached in the dead time of the machine (~0.1 s), and the height of the peak for $PP_i$ is about twice that of IDP (Fig. 8). Consequently, the rate of ion-pumping is at least 12–25 times that of $PP_i$ hydrolysis/phosphate release. The decay constants (protein unable to pump) for the two are similar (Table 3), suggesting that they correspond to a similar event. We thus hypothesise that $PP_i$ hydrolysis/phosphate release from the liposomes at 20 °C is rate limiting. Indeed, we observed that to fully recover the signal on the same sensor, a waiting time of several minutes was required, in line with the proposition that hydrolysis or phosphate release is slow. The logical explanation for the twice-as-big signal from $PP_i$ as compared to IDP is that IDP leads to a single- pumping event, whereas $PP_i$ triggers two pumping events. In other words, with $PP_i$ the enzyme can bind and pump two $Na^+$ within a catalytic cycle (one from each subunit, with at least a single turnover in subunit A) while non-hydrolysable IDP only allows the binding and pumping of one $Na^+$ (Fig. 9). In the context of half-of-the-sites reactivity, this means that ion and substrate binding to subunit B cannot take place until hydrolysis happens in subunit A—and no hydrolysis can happen with IDP. Similar results have been reported for a $K^+$-dependent $H^+$-PPase in which $PP_i$ produced a ~tenfold bigger signal (multiple turnovers) than other non-hydrolysable $PP_i$ mimics (single-pumping event upon binding of one molecule) (Li et al, 2016; Shah et al, 2017). The measured currents in the presence of IDP are specifically caused by a single ion-pumping event and not by $Mg^{2+}$ being brought slightly beneath or close to the membrane surface as $PP_i$ mimics that allow $loop_{5\text{-}6}$ closure and thus pumping (e.g., IDP) induce a bigger signal than those that do not (e.g., etidronate), and these signals can be collapsed by ionophores specific for the pumped ion (Li et al, 2016; Shah et al, 2017). Moreover, $PP_i$ (or

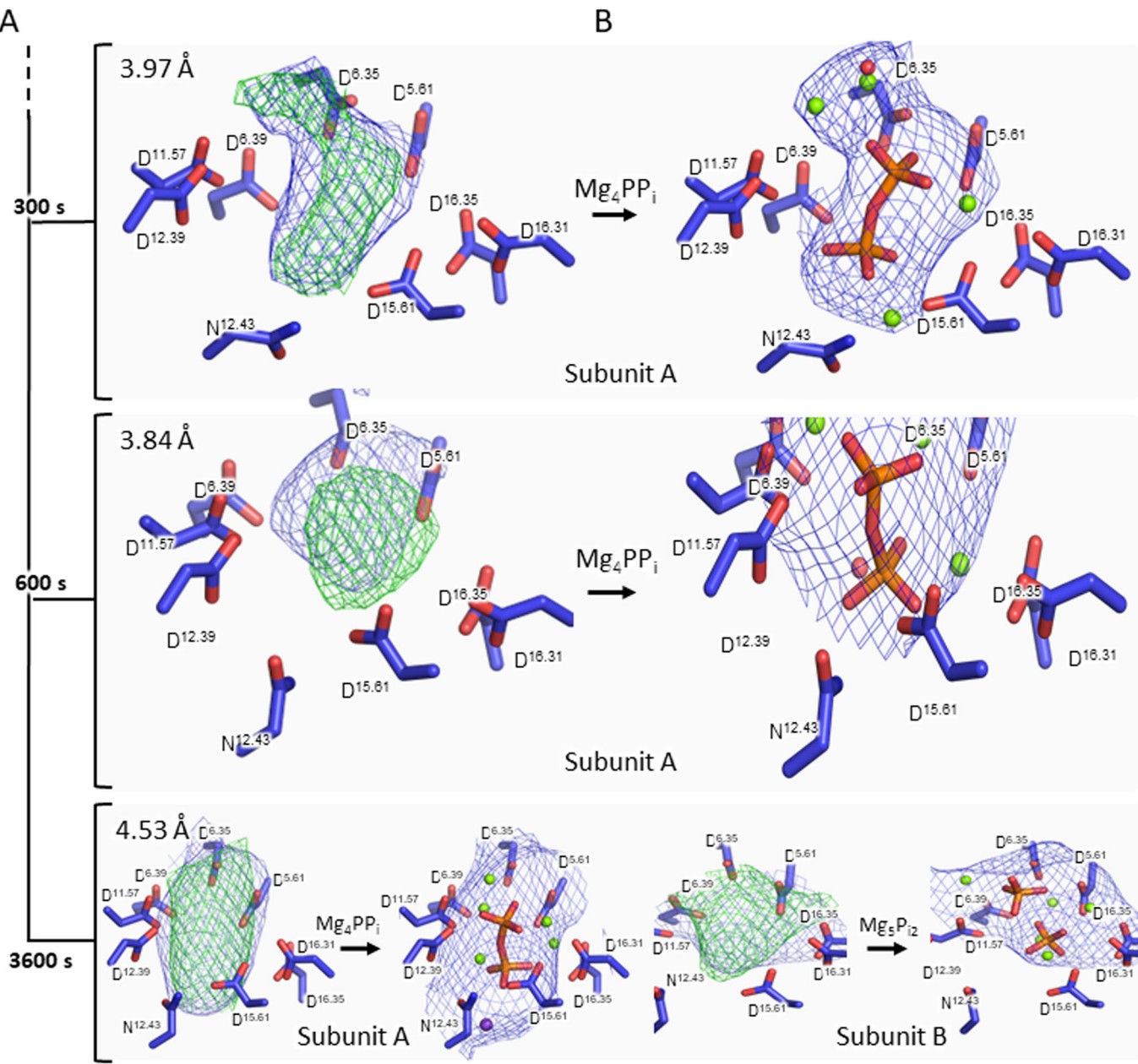

**Figure 7. Active site occupation in time-resolved *Tm*PPase structures.**

The active sites of asymmetrically occupied subunit A (300 s and 600 s) and "symmetrically" occupied subunit A and B (3600 s) are shown with the 2mF$_o$–dF$_c$ density (blue) and mF$_o$–dF$_c$ density (red/green) for ligand before and after modelling. The 2mF$_o$–dF$_c$ density is contoured at 1 σ (0–60 s -600 s) or 2 σ (3600 s) and the mF$_o$–dF$_c$ density is contoured at 3 σ. The resolution of the combined structures at each time point is stated. Bound ions and water are displayed as spheres (Mg$^{2+}$: green; K$^+$: purple; water: red). (A) 2mF$_o$-dF$_c$ density and mF$_o$–dF$_c$ density for ligand shown before modelling. (B) 2mF$_o$-dF$_c$ density and mF$_o$–dF$_c$ density for ligand shown after modelling.

IDP) most likely enter as Mg$_2$-complex under assay conditions (Maeshima, 2000; Malinen et al, 2008), which is electro-neutral and thus cannot trigger a signal. Consequently, pumping must precede hydrolysis.

Conversely, two recently published papers from Baykov and co-workers (Baykov et al, 2022; Malinen et al, 2022) continue to posit their billiard-type mechanism, where hydrolysis precedes pumping. By using quenched-flow measurements on *Tm*PPase at 40 °C, they demonstrate that hydrolysis is the most likely rate-determining

step, which is consistent with our electrometric measurements on *Tm*PPase at 20 °C. They propose that this means that hydrolysis occurs at the same time or precedes pumping—i.e., the chemical proton released from hydrolysis enters an ion pathway and forces the pumped Na$^+$ into the exit channel. However, their pre-steady-state rate at 40 °C is 12 s$^{-1}$ (Malinen et al, 2022; Fig. 2), consistent with the $k_{cat}$ of our steady-state kinetics at 20 °C (0.2–0.4 s$^{-1}$) (Appendix Table S7). The pre-steady-state rate at 20 °C is thus likely about 1–3 s$^{-1}$ (a factor of 2–3 per 10 °C)

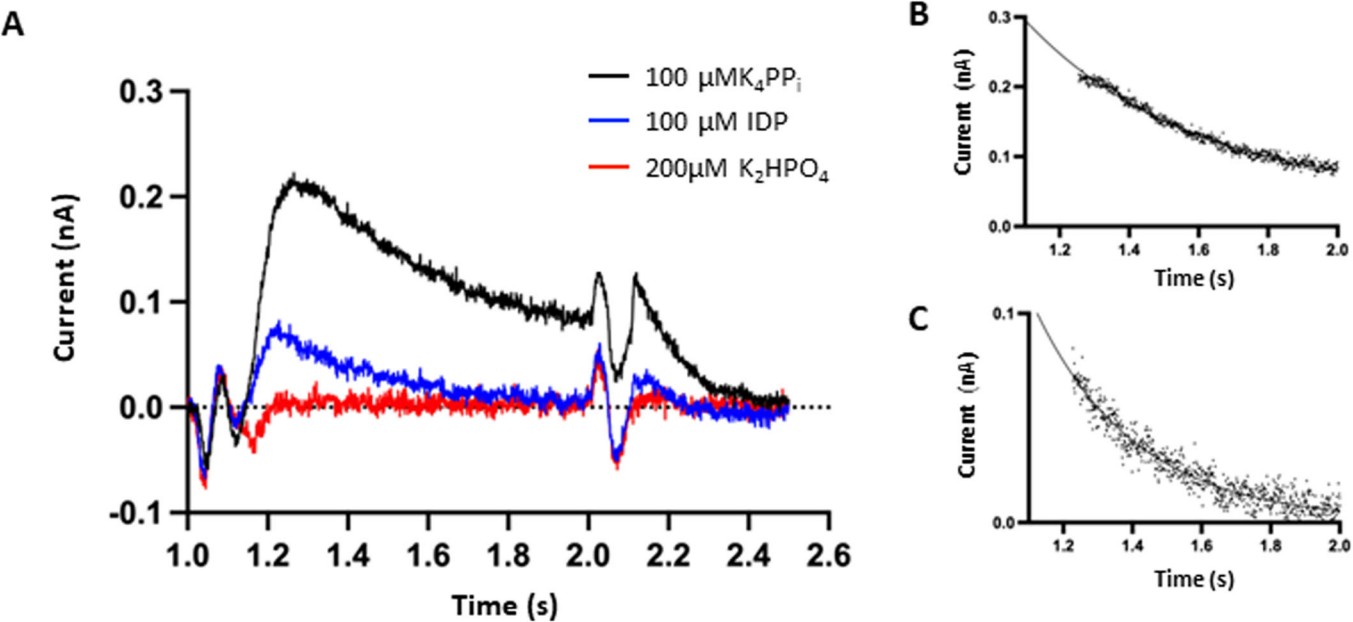

**Figure 8. Transient currents of *Tm*PPase Na⁺ pumping.**

(A) Pumping triggered by 100 μM of K₄PPi, 100 μM of IDP and 200 μM of K₂HPO₄. (B) Current exponential decay fit curve of PPᵢ (1.2–2 s). (C) Current exponential decay fit curve of IDP (1.2–2 s). Source data are available online for this figure.

**Table 3. Current decay parameters for Na⁺ pumping of *Tm*PPase.**

| Parameter | Value | |
|---|---|---|
| | 100 μM PPᵢ | 100 μM IDP |
| Equation | Y = (Y0-Plateau) *exp(-k*X) +Plateau | Y = (Y0-Plateau) *exp(-k*X)+Plateau |
| Y0 (nA) | 2.569 | 4.752 |
| Plateau (nA) | 0.04111 | 0.0007791 |
| $k$ (s⁻¹) | 2.087 | 3.441 |
| Half-time (s) | 0.3321 | 0.2014 |
| Tau (s) | 0.4791 | 0.2906 |
| Span (nA) | 2.528 | 4.751 |
| Degrees of freedom | 742 | 770 |
| R squared | 0.9822 | 0.9235 |
| Sum of squares | 0.02275 | 0.02009 |
| Sy.x | 0.005537 | 0.005107 |

(Blehrádek, 1926)—i.e., at least 4–10 times slower than the rates observed in the electrometric measurements, where PPᵢ binding and ion transfer happens within the dead time of the machine and the decay following the initial pumping event is of similar speed. Again, the simplest explanation for this is that pumping precedes hydrolysis: the pumping of Na⁺ increases the negative charge in the closed active site, causing deprotonation along the ion pathway and hydrolysis of the PPᵢ. We concur with Baykov and co-workers that, whatever the mechanism, it must be the same for all M-PPases as the catalytic machinery is so similar. We also agree that the most

likely identity of the charged residue, as identified by their steady-state solvent isotope experiments, is indeed the semi-conserved glutamate (E^{6.53/57}) with a $pK_a$ of about 7.8. A glutamate in the membrane that has just lost a counter-ion would suit perfectly.

## Ion selectivity in M-PPases

Finally, we can extend the model of half-of-the-sites reactivity and pumping-before-hydrolysis to also explain ion selectivity. The current model of ion selectivity (Li et al, 2016) posits that M-PPases with E^{6.53} are Na⁺-pumps and ones with E^{6.57} are H⁺-pumps (Nordbo et al, 2016)—but both Na⁺,H⁺-PPases and K⁺-independent H⁺-PPases have E^{6.53} (Table 1). What explains this?

Our new proposal is that ion selectivity depends on the interaction between the semi-conserved glutamate with a highly conserved serine on helix 5, S^{5.43} (pairwise identity: 90.6%). This interaction, propagated through the dimer interface via the adjacent helix 13, mediates helix 5 orientation and ion selectivity/binding. The ion gate configuration is thus directly linked to the orientation of helix 5, which determines whether E^{6.53} points towards S^{5.43} (a proton pumper), or away from S^{5.43} (a Na⁺-pumper) (Movie EV1). In what follows, we show that this proposal convincingly accounts for all currently known M-PPase subclasses (Fig. 10).

In K⁺-independent H⁺-PPases, protonated E^{6.53} is held in place by S^{5.43} (Fig. 10A), while in Na⁺-PPases, the bent helix 5 forces E^{6.53} to point away from S^{5.43} to avoid clashing, creating a Na⁺-binding site (Fig. 10B). Our *Tm*PPase structure obtained in the absence of Na⁺ (0–60-s structure) suggests that in low Na⁺ conditions, E^{6.53} is not tightly coordinated at the ion gate, allowing it to loosely interact with S^{5.43}. This could thus explain why K⁺-dependent Na⁺-PPases can pump protons at Na⁺ concentrations <5 mM (Luoto et al, 2013). In

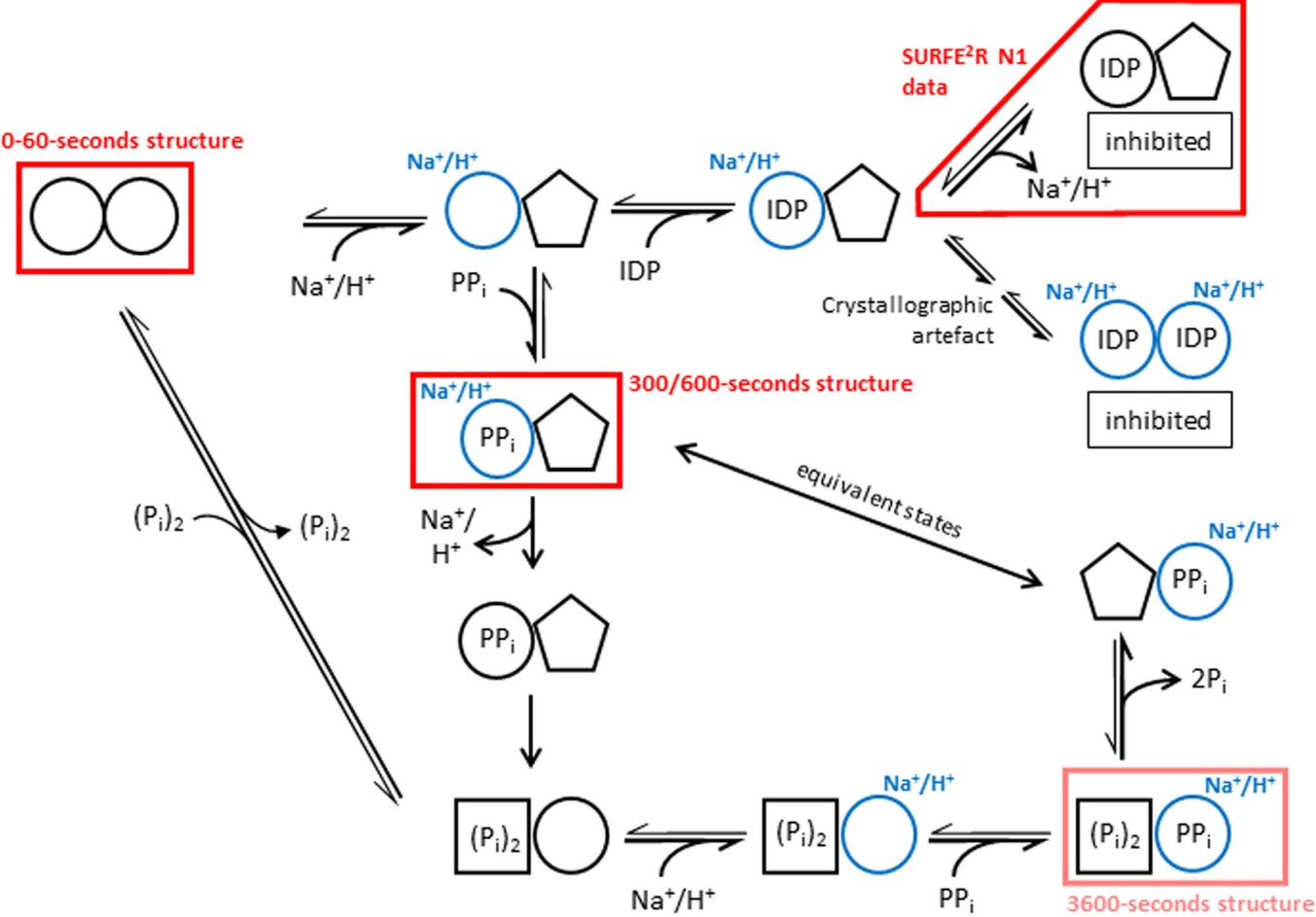

**Figure 9. Unified model of M-PPase catalysis.**

Schematic model unifying functional asymmetry (half-of-the-sites-reactivity) and energy coupling (pumping-before-hydrolysis). The active site status is defined by shape, and binding of the pumped ion at the active side, indicated by a label and blue colouring. $PP_i$ can only bind to one subunit at a time and requires prior binding of the pumped ion at the ion gate. The thermodynamically favoured reaction pathway is indicated by bold arrows, with the rate-determining step shown as a single arrow. Red boxes highlight enzyme states mapped by time-resolved crystallography or investigated in electrometric studies.

K⁺-dependent H⁺-PPases, $E^{6.57}$ can be coordinated by $S^{5.43}$ independent of helix 5 geometry (Fig. 10C), imposing a strict H⁺-selectivity. The unusual *Flavobacterium johnsoniae* (Luoto et al, 2011) H⁺-PPase, where the semi-conserved glutamate is at position 5.43, is also consistent with our model. The side-chain carboxylate of $E^{5.43}$ cannot promote Na⁺-binding about 5 Å away, but can be modelled to occupy a position similar to that of $E^{6.57}$ in *Vr*PPase:Mg₅IDP, allowing protonation and deprotonation of $E^{5.43}$ (Tsai et al, 2014).

To explain ion-pumping selectivity in K⁺-dependent Na⁺,H⁺-PPases, we posit that the ion gate configuration flips between the H⁺-PPase (*Pa*PPase:Mg₅IDP) and Na⁺-PPase (*Tm*PPase:Mg₅IDP) conformations. For instance, if subunit A is in the H⁺-pumping configuration, conformational changes through the helix 5–13 connection convert subunit B into the Na⁺-binding conformation and vice versa (Fig. 10D). Seen this way, the K⁺-dependent Na⁺,H⁺-PPases use a variant of the standard M-PPase mechanism, rather than being sui generis. To confirm this, the complete catalytic cycle must be mapped in time-resolved structural studies at high enough resolution to resolve individual side-chain orientations. We also suggest that *all* M-PPases, not just the few studied so far (Anashkin

et al, 2021; Artukka et al, 2018; Vidilaseris et al, 2019), will show half-of-the-sites reactivity.

## A unified principle of catalysis

In summary, the only sequence of events that is consistent with all the data presented above, including the experiments by Baykov and co-workers (Malinen et al, 2022), is based on a half-of-the-sites reactivity mechanism, supports a pumping-before-hydrolysis type energy coupling and is as follows (Fig. 9): *Productive* binding of substrate in monomer A requires binding of an ion at the gate (though non-productive modes like those in Fig. 6C—300 s may occur in the absence of ion (Malinen et al, 2008)). Following *productive* substrate binding, the downward motion of helix 12, the helical rearrangements at helices 6 and 16 and the straightening and tighter winding of helix 5 trigger ion-pumping. The structural rearrangements are propagated into subunit B via helices 5, 10, 12, 13 and the exit channel loops as pointed out above where they prevent binding of an ion or a substrate at monomer B. Pumping in monomer A drives the hydrolysis in monomer A, which then

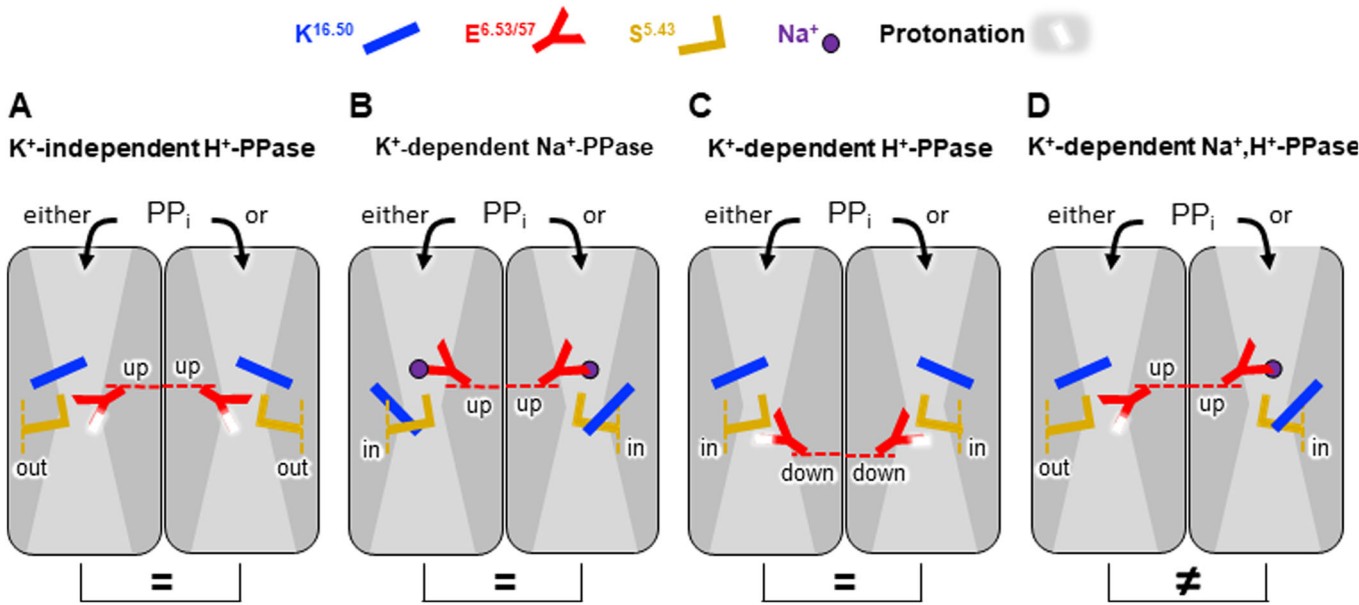

**Figure 10. Schematic model of ion-pumping selectivity in M-PPases.**

The arrangement of $K^{16.50}$ (|) $E^{6.53/57}$ (Y) and $S^{5.43}$ (L) at the ion gate defines ion-pumping across all M-PPase subclasses. The ion gate set-up is shown with both subunits in the *same* catalytic state for comparison. In single-pumping M-PPases, the ion gate conformation of subunits A and B are the same (**A–C**) when set up for pumping. In dual-pumping M-PPases (**D**), the ion gate conformation of subunit A and B are asymmetrical when set up for pumping. The position of helix 5 (in: $K^+$-dependent M-PPases/out: $K^+$-independent M-PPases) and the semi-conserved glutamate (up: $E^{6.53}$/down: $E^{6.57}$) are indicated by dashed vertical and horizontal lines, respectively. (**A**) $E^{6.53}$ protonation (white glow) and interaction with $S^{5.43}$. $K^{16.50}$ destroys the $Na^+$-binding site. (**B**) $Na^+$ (purple sphere) binds to the ion gate and interacts with $E^{6.53}$, which faces away from $S^{5.43}$ because helix 5 is too close. (**C**) $E^{6.57}$ protonation (white glow) because of its shift one helix turn down, which creates sufficient space for the $S^{5.43}$-$E^{6.53}$ interaction. (**D**) Functional asymmetry in $K^+$-dependent $Na^+$, $H^+$-PPases, explaining $Na^+$ binding and $E^{6.53}$ protonation based on helix 5 orientation and $E^{6.53}$-$S^{5.43}$ interaction. Subunit A is similar to (**B**), whereas subunit B is similar to (**C**). The binding of $PP_i$ to monomer A leads to proton pumping, while binding of $PP_i$ to monomer B leads to $Na^+$ pumping.

releases monomer B into an ion-binding/substrate binding conformation. Afterwards, ion binding in monomer B could allow release of product in monomer A, followed by substrate binding and pumping in B. The beauty of this mechanism is that it provides a convincing rationale for all the observed data, in particular explaining half-of-the-sites reactivity and dual-pumping M-PPases. It does not require that pumping is the rate-determining step, and indeed is consistent with it not being so. Other models, positing that two ions can occasionally be pumped in one subunit are, to our mind, not as convincing as no modern experiments on purified proteins have indicated a hydrolysis/transport ratio above 1:1.

Our model provides, for the first time, an overall explanation of ion selectivity and catalysis in *all* M-PPases and makes testable predictions: e.g., global conformational changes should occur in the first 0.1–0.2 s at 20 °C. These need to be tested through functional and structural studies, in particular the use of time-resolved, single-molecule techniques to capture further details of mechanism, as well as molecular dynamics simulations—efforts that are already underway (Holmes et al, 2022).

## Methods

### Mutagenesis of *Pa*PPase

We used N-terminally $RSGH_6$-tagged constructs for full-length *Tm*PPase expression from Kellosalo et al, 2011 and replaced the open reading frame encoding for *Tm*PPase with a section encoding for full-length *Pa*PPase instead. The *Pa*PPase gene was PCR amplified (Q5® Hot Start High-Fidelity 2X Master Mix from NEB, Frankfurt am Main, Germany) with primers introducing a GG-linker along with a 5'-SalI (TTT TTT GTC GAC ATG CAT CAC CAT CAC CAT CAC GGT GGA AAT ATG ATA AGC TAT GCC TTA CTA GG) and 3'-XbaI (TTT TTT TCT AGA TCA GAA AGG CAA TAG ACC TG) restriction site. The PCR product was inserted into the linearised (SalI, XbaI from NEB) pRS1024 yeast expression vector (Kellosalo et al, 2011). *Pa*PPase variants $K^{12.46}A$ (C AAT ACC ACA gca GCC ACT ACT AAG GG, CC GAC GGA GTC CAG TAC A), $T^{12.49}A$ (A AAA GCC ACT gct AAG GGA TAT GC, GT GGT ATT GCC GAC GGA G) and the combination of both were generated using the Q5® site-directed mutagenesis kit (NEB) (lower case letters highlight the amino acid change). Template DNA was removed by DpnI (NEB) digestion, and the constructs were sequenced to confirm the introduction of point mutations.

### Protein expression and purification

We expressed and purified *Pa*PPase and *Tm*PPase in *Saccharomyces cerevisiae* as described elsewhere (López-Marqués et al, 2005; Strauss et al, 2018). In brief, yeast expression plasmids carrying N-terminally 6xHis-tagged wild-type or variant *Pa/Tm*PPase under control of the constitutively active PMA1 promoter were freshly transformed into the *S. cerevisiae* strain BJ1991 (genotype: *MATα prb1-1122 pep4-3 leu2 trp1 ura3-52 gal2*) and

cultivated at 30 °C for 12 h in 250 mL selective synthetic complete dropout starter cultures lacking leucine (SCD-Leu, in-house). In total, 750 mL of 1.5× SCD-Leu (*Pa*PPase) or 1.5× yeast peptone dextrose (YPD, in-house)(*Tm*PPase) expression culture were inoculated with 250 mL of starter culture for protein expression at 30 °C. Cells were harvested after 8–10 h from 10 L of expression batches by centrifugation (4000×*g*, 4 °C, 15 min).

Cells were lysed using a bead-beater (Biospec Products, Bartlesville, Oklahoma) with 0.2-mm glass beads and membranes were collected by ultracentrifugation (100,000×*g*, 4 °C, 1 h). The membrane pellet was resuspended in 50 mM MES pH 6.5, 20% v/v glycerol, 50 mM KCl, 5 mM $MgCl_2$, 2 mM dithiothreitol (DTT), 1 mM phenylmethylsulfonyl fluoride (PMSF) and 2 µg/mL pepstatin to a final total protein concentration of ~7 mg/mL, mixed with solubilisation buffer (50 mM MES-NaOH pH 6.5 20% v/v glycerol, 5.34% w/v n-dodecyl-β-D-maltoside (DDM), 1 mM $K_4PP_i$) at a 3:1 ratio and incubated at 75 °C ("hot-solve") for 1.5 h. Protein was then purified by IMAC using nickel-NTA resin (Bio-Rad, Hercules, California). Depending on the protein and downstream experiments, different buffers were used for purification as outlined below. For structural studies of *Pa*PPase, the solubilised membranes were incubated with nickel-NTA (Cytiva, Marlborough, MA) resin at 40 °C for 1–2 h and washed with 2 column volumes (CV) 50 mM MES-NaOH pH 6.5, 20% v/v glycerol, 5 mM $MgCl_2$, 20 mM imidazole, 1 mM DTT and 0.5% w/v n-decyl-β-D-maltoside (DM) or 0.05% w/v DDM prior to elution in 2 CV 50 mM MES-NaOH, pH 6.5, 3.5% v/v glycerol, 5 mM $MgCl_2$, 400 mM imidazole, 1 mM DTT and 0.5% w/v DM or 0.05% DDM w/v. *Pa*PPase samples used in functional studies were solubilised in DDM and contained not MES-NaOH but MOPS-TMAOH (tetramethylammonium hydroxide) (pH 6.5) instead in order to obtain a "$Na^+$-free" sample. The purification of *Tm*PPase followed a similar protocol with 0.5% w/v octyl glucose neopentyl glycol (OGNG) and MES-TMAOH (pH 6.5). In addition, the purification buffers contained 50 mM KCl due to the $K^+$-dependence of *Tm*PPase. After nickel-NTA purification, all purified proteins were exchanged into elution buffer lacking imidazole using a PD10 desalting column (Cytiva) and concentrated to ~10 mg mL$^{-1}$. SDS-PAGE and size exclusion chromatography (SEC) using a Superose® 6 Increase 10/300 GL column (Cytiva) and an NGC Quest 10 Plus System (Bio-Rad) showed that both wild-type and variant proteins were pure and monodisperse (Appendix Fig. S1).

## Vapour diffusion crystallisation of *Pa*PPase and *Tm*PPase

Initial crystallisation trials of wild-type *Pa*PPase were carried out with several commercial sparse matrix screens using protein solubilised in DM and DDM. Commercial sparse matrix crystallisation screens were set up with protein at 10 mg mL$^{-1}$ (1:1 ratio) after pre-incubation with 4 mM $Na_4IDP$ (imidodiphosphate) salt or $CaCl_2$ (1 h, 4 °C). Any precipitation that formed within the incubation period was removed by centrifugation at 10,000×*g* for 10 min prior to setting up crystallisation trials. The best crystals were obtained in the presence of 2 mM IDP in 30–33% v/v PEG 400, 0.1 M MES pH 6.5, 0.05 M $LiSO_4$, and 0.05 M NaCl at 20 °C with protein solubilised in DM. The crystals were manually harvested at 20 °C. X-ray diffraction was improved by keeping the harvested crystal in the loop for 10 s prior to flash cooling in liquid nitrogen, which effectively led to crystal dehydration.

Initial vapour diffusion crystallisation trials of wild-type *Tm*PPase were based on a published crystallisation condition (36% v/v PEG 400, 100 mM Tris-HCl pH 8.5, 100 mM $MgCl_2$, 100 mM NaCl, 2 mM DTT) (Li et al, 2016) that was further optimised for time-resolved experiments (i.e., to contain no inhibitors). *Tm*PPase in OGNG was set up at 10 mg mL$^{-1}$ after pre-incubation with 0.4–4.0 mM $K_4PP_i$ (1 h, 4 °C) (instead of $Na_4IDP$) and all crystallisation buffers had NaCl replaced with KCl. The best crystals formed in 24–26% v/v PEG 400, 50–60 mM Tris-HCl pH 8.5, 2–3 mM $MgCl_2$, 175 mM KCl, 2 mM DTT and 0.4 mM $K_4PP_i$ (1:1 ratio) at 20 °C.

## *Pa*PPase data collection, structure solution and refinement

*Pa*PPase crystals were sent to several beamlines including I04 and I24 at the Diamond Light Source (DLS) and ID23-1 and MASSIF-1 at the European Synchrotron Radiation Facility (ESRF) for data collection at 100 K. Collected datasets were processed in XDS (Kabsch, 2010) and the structure was solved by molecular replacement in Phaser (McCoy et al, 2007) using a homology search model based on the 3.5 Å structure of *Tm*PPase:$Mg_5IDP$ (protein data bank (PDB) ID: 5LZQ) with loop regions removed. The crystals were extremely radiation sensitive, so a complete data set could not be collected on any of them. Consequently, the first few hundred images of eight datasets (3.84–4.35 Å) with positive density for $Mg_5IDP$ in the active site, less than 2% deviation in unit cell parameters and identical space group ($P2_1$) were combined in XDS using XSCALE (Kabsch, 2010). The combined dataset was submitted to the STARANISO webserver (Tickle et al, 2018) prior to molecular replacement. Several rounds of refinement using Phenix. refine (Liebschner et al, 2019) and manual modelling in Coot (Emsley et al, 2010) were carried out. After an initial round of rigid-body refinement with grouped B-factors, tight restraints including torsion angle, non-crystallographic symmetry (NCS), secondary structure, and reference structure (PDB: 4A01) restraints were applied to maintain a realistic geometry. In the last rounds of refinement, Translation–Libration–Screw–rotation (TLS) was enabled, and restraints were released except for torsion angle NCS restraints, which were retained to prevent overfitting.

## Time-resolved cryo-trapping X-ray crystallography and structure solution

Time-resolved cryo-trapping crystallography experiments on *Tm*PPase were conducted by manual soaking of vapour diffusion crystals grown in the absence of $Na^+$ but in the presence of $PP_i$ in a $Na^+$-containing trigger solution (60 mM Tris-HCl pH 8.0, 26% v/v PEG 400, 175 mM KCl, 2.4 mM $MgCl_2$, 2 mM $K_4PP_i$, 20 mM NaCl) to initiate the enzymatic reaction in crystallo. The reaction was stopped by flash cooling in liquid nitrogen after different soaking times ($t = 0$ [no $Na^+$ applied], 1, 2, 5, 10, 60, 300, 600, 3600 s) that were selected based on the $k_{cat}$ of *Tm*PPase under conditions similar to the crystallisation conditions (Appendix Table S7). Crystallisation wells were re-sealed if the soaking time exceeded 60 s to minimise evaporation. Up to five crystals were used for each time point.

Diffraction data were collected at 100 K at beamline P14-I at the Deutsches Elektronen Synchrotron (DESY), and the data were

processed in XDS (Kabsch, 2010) or Xia2/DIALS (Winter et al, 2018). This was followed by anisotropy correction using the STARANISO webserver (Tickle et al, 2018) and molecular replacement in Phaser (McCoy et al, 2007) using the *Tm*PPase:-CaMg (PDB: 4AV3) structure without heteroatoms as a search model. The similarity between unit cells of the collected datasets was analysed in BLEND (Foadi et al, 2013) and datasets of the same or different time points ($t = 0$–60 s) were combined if the linear cell variation was below 3% and the space group and active site status (occupied versus not-occupied) were identical.

The single best non-activated structure (reference) and the grouped $t = 0$–60 s structure (subset of cluster A) were subject to several rounds of refinement using phenix.refine (Liebschner et al, 2019) and manual modelling in Coot (Emsley et al, 2010). After an initial round of rigid-body refinement with grouped B-factors, torsion angle NCS restraints were applied to further reduce the number of parameters in refinement alongside optimised X-ray/B-factor and X-ray/stereochemistry weighting by phenix. In the final refinement rounds, TLS was applied as well. The 0–60-s structure of *Tm*PPase was then used as a search model for molecular replacement of combined datasets that were collected at longer delays after Na$^+$-activation. Refinement of these data followed a similar protocol, but the 300-s dataset was limited to a single round of 5 refinement cycles, which was sufficient to check for changes of the overall helix geometry at the active site or ion gate. Additional secondary structure restraints were applied in the refinement of the low-resolution $t = 600$ s and $t = 3600$ s *Tm*PPase structures to maintain realistic geometry.

## Structure analysis

Geneious R11 was used to search the UniProtKB/Swiss-Prot database with blastp (Altschul et al, 1990) for similar sequences to *Pa*PPase and the results were aligned using the Geneious global alignment tool with free end gaps to determine residue conservation and sequence identity. Structure alignments and the r.m.s.d. calculations were done in PyMol 2.2.3 (Schrödinger, LLC, 2015). The standard deviation was stated when multiple structures were compared by their r.m.s.d.. The solvent-accessible surface areas and volumes were determined using HOLLOW with a 1.4–1.5 Å interior probe size (Ho and Gruswitz, 2008). Inter-atom difference distance matrices (DiDiMa) of Cα atoms were generated by the Bio3D R-package for structural bioinformatics (Grant et al, 2006). Hydrogen-bonding patterns were analysed in HBplus using default settings (McDonald and Thornton, 1994). The local (residue by residue) helix curvature analysis was done considering blocks of four residues using the Bendix plugin of the Visual Molecular Dynamics suite (Dahl et al, 2012), whereas the global (helix by helix) curvature analysis was done using the HELANAL-Plus webserver (Kumar and Bansal, 2012).

## Fixed-time P$_i$-release assay for activity measurements under crystallisation conditions

The hydrolytic activity of purified *Tm*PPase for time-resolved structural studies was assessed by using the molybdenum blue reaction method with relipidated (12 mg mL$^{-1}$ L-α-lecithin) protein in DDM:OGNG mixed micelles as previously described (Baykov et al, 2021; Strauss et al, 2018). The reaction buffers were matched

to the crystallisation conditions in order to estimate time scales of substrate turnover in crystallo. The concentration of MgCl$_2$ and K$_4$PP$_i$ required to maintain 5 mM free Mg$^{2+}$ at pH 8.0 was approximated as described by Baykov and co-workers (Baykov et al, 1993). As reference, a routine reaction was done in 60 mM Tris-HCl pH 8.0, 5 mM free Mg$^{2+}$, 100 mM KCl, and 20 mM NaCl at 71 °C for 5 min. Subsequent reactions at 20 °C were incubated for 240 min instead as this produces detectable reaction product in a linear range. The activity of protein in the optimised vapour diffusion crystallisation condition (60 mM Tris-HCl pH 8.0, 100 mM KCl, 3 mM MgCl$_2$, 175 mM KCl, 26% v/v PEG 400, 400 μM K$_4$PP$_i$) was tested upon reaction initiation with 20 mM NaCl with and without relipidated sample. The standard error of the mean (SEM) was obtained from three technical repeats.

## Continuous-flow P$_i$-release assay

Kinetic experiments for wild-type and variant *Pa*PPase were done using a phosphate analyzer (Baykov et al, 2021) with relipidated (12 mg mL$^{-1}$ L-α-lecithin) protein in DDM micelles. The reaction mixture of 40 mL contained 50 mM MOPS-TMAOH buffer (pH 7.2) and varying concentrations of free Mg$^{2+}$ (added as MgCl$_2$) and TMA$_4$PP$_i$ in ratios that gave the desired concentration of Mg$_2$PP$_i$ as substrate (Baykov et al, 1993). Reactions were initiated by protein (at low substrate concentration) or TMA$_4$PP$_i$ (at high substrate concentration) and the P$_i$ accumulation was continuously recorded for 2–3 min at 40 °C. Reaction rates were calculated from the initial slopes of the P$_i$ liberation and analysed using Prism 6.0 (GraphPad Software) based on a model assuming allosteric substrate binding in dimeric enzyme (Eq. (1); Appendix Fig. S6) (Anashkin et al, 2021) or a standard Michaelis–Menten type mechanism.

$$\boldsymbol{v} = (\boldsymbol{V_1} + \boldsymbol{V_2}[\mathbf{S}]/\boldsymbol{K_{M2}})/(1 + \boldsymbol{K_{M1}}/[\mathbf{S}] + [\mathbf{S}]/\boldsymbol{K_{M2}}) \qquad (1)$$

## Nanion SURFE$^2$R N1 measurement

For the Nanion SURFE$^2$R N1 experiments, purified *Tm*PPase was reconstituted into liposomes as previously described (Li et al, 2016) with some modifications. Briefly, the purified protein was buffer exchanged into a reconstitution buffer (50 mM MOPS-KOH pH 7.2, 50 mM KCl, 5 mM MgCl$_2$, 2 mM DTT) to remove Na$^+$ and glycerol and then diluted to 50 μg mL$^{-1}$. In total, 120 mg of soybean lectin was dissolved in 1 mL of water and tip sonicated with 60% amplitude, 6 s pulses for 1 min with 1 min on ice between sonications until the solution was clear. In total, 15 μL liposomes solution (120 mg mL$^{-1}$ soybean lecithin in 50 mM MOPS-KOH pH 7.2) was mixed with 1 mL of diluted protein sample. SM-2 Bio-beads were added in small increments to a final concentration of 0.25 mg μL$^{-1}$ and then placed into a mixer at 4 °C for 6 h to ensure that the beads stayed in suspension. The proteoliposomes were collected and frozen at −80 °C in aliquots. To ensure that the reconstituted protein was still active, the hydrolytic activity was assessed in fixed-time P$_i$-release assays as described above.

Electrometric measurements were performed on the SURFE$^2$R N1 instrument (Nanion Technology). The gold sensors were prepared based on the SURFE$^2$R N1 protocol. This involves thiolating the gold sensor surface and covering it with a lipid layer using sensor prep A2 and B solutions. The resulting solid support

membrane-based biosensor can be used to immobilise liposomes containing *Tm*PPase. Overall, 15 μL of sonicated proteoliposomes followed by 50 μL of *Tm*PPase SURFE²R N1 buffer (50 mM MOPS-KOH pH 7.2, 50 mM NaCl, 5 mM $MgCl_2$) were applied directly to the sensor surface. Sensors were centrifuged for 30 min at 2000×*g* and incubated at 4 °C for 3 h. After mounting the sensors in the SURFE²R N1, the sensors were rinsed once with 1 mL rinsing buffer (50 mM MOPS-KOH pH 7.2, 50 mM NaCl, 5 mM $MgCl_2$). Measurements were performed for 3 s by consecutively flowing non-activating buffer B (50 mM MOPS-KOH 7.2, 50 mM NaCl, 5 mM $MgCl_2$, 200 μM $K_2HPO_4$) and activating buffer A (50 mM MOPS-KOH, 50 mM NaCl, 5 mM $MgCl_2$, $K_4PP_i$ or IDP) across the sensor for 1 s each in a BAB sequence. Thus, charge transport across the membrane is initiated by $K_4PP_i$ or IDP in buffer A, which is flowed across the sensor during the time period between 1 and 2 s. Transport of positively charged ions during this time results in a positive electrical current, which is the signal output of the SURFE²R N1 instrument. Between each measurement, the sensor was washed with 1 mL rinsing buffer and incubated for 60 s. Both substrates $K_4PP_i$ and IDP were tested in triplicates.

## Data availability

The atomic coordinates and structure factors of the *Pa*PPase:Mg₅IDP complex (PDB ID: 8B37) and the grouped/combined time-resolved *Tm*PPase structures at 0–60-seconds (PDB ID: 8B21), 300-seconds (PDB ID: 8B22), 600-seconds (PDB ID: 8B23) and 3600-seconds (PDB ID: 8B24) have been deposited in the PDB.

## Peer review information

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

## Acknowledgements

JS acknowledges funding from the European Union's Horizon 2020 research and innovation programme under the Marie Skłodowska-Curie grant 722687. CW was supported by the Leeds 110th Anniversary Research Scholarships. AG and LJ acknowledge funding from the BBSRC (grant: BB/T006048/1). AG, KV and AMM received funding from the Academy of Finland (grants: 1322609, 308105, and 307775). ARP was supported by the Cluster of Excellence "The Hamburg Centre for Ultrafast Imaging" and "CUI: Advanced Imaging of Matter" of the Deutsche Forschungsgemeinschaft (DFG EXC1074, EXC2056). T-REXX is supported by the Bundesministerium für Bildung und Forschung ("Verbundforschung", 05K16GU1 and 05K19GU1). We thank the Leeds Astbury Centre for Molecular and Structural Biology for support, Diamond Light Source for access to beamline I04 and I24, the European Synchrotron Radiation Facility for access to beamline ID23-1 and MASSIF-1, and EMBL for access to beamline P14.I and P14.II (T-REXX) at PETRA III (mx747, mx839, mx862).

## Author contributions

**Jannik Strauss**: Conceptualisation; Formal analysis; Investigation; Visualisation; Methodology; Writing—original draft; Project administration; Writing—review and editing. **Craig Wilkinson**: Conceptualisation; Investigation. **Keni Vidilaseris**:

Formal analysis; Investigation; Visualisation; Writing—original draft; Writing—review and editing. **Orquidea M De Castro Ribeiro**: Investigation. **Jianing Liu**: Investigation. **James Hillier**: Investigation. **Maximilian Wichert**: Investigation. **Anssi M Malinen**: Resources; Formal analysis; Investigation. **Bernadette Gehl**: Investigation. **Lars JC Jeuken**: Resources; Supervision; Funding acquisition; Writing—review and editing. **Arwen R Pearson**: Conceptualisation; Resources; Supervision; Funding acquisition; Methodology; Writing—review and editing. **Adrian Goldman**: Conceptualisation; Formal analysis; Supervision; Funding acquisition; Writing—original draft; Project administration; Writing—review and editing.

## Disclosure and competing interests statement

The authors declare no competing interests.

# Expanded View Figures

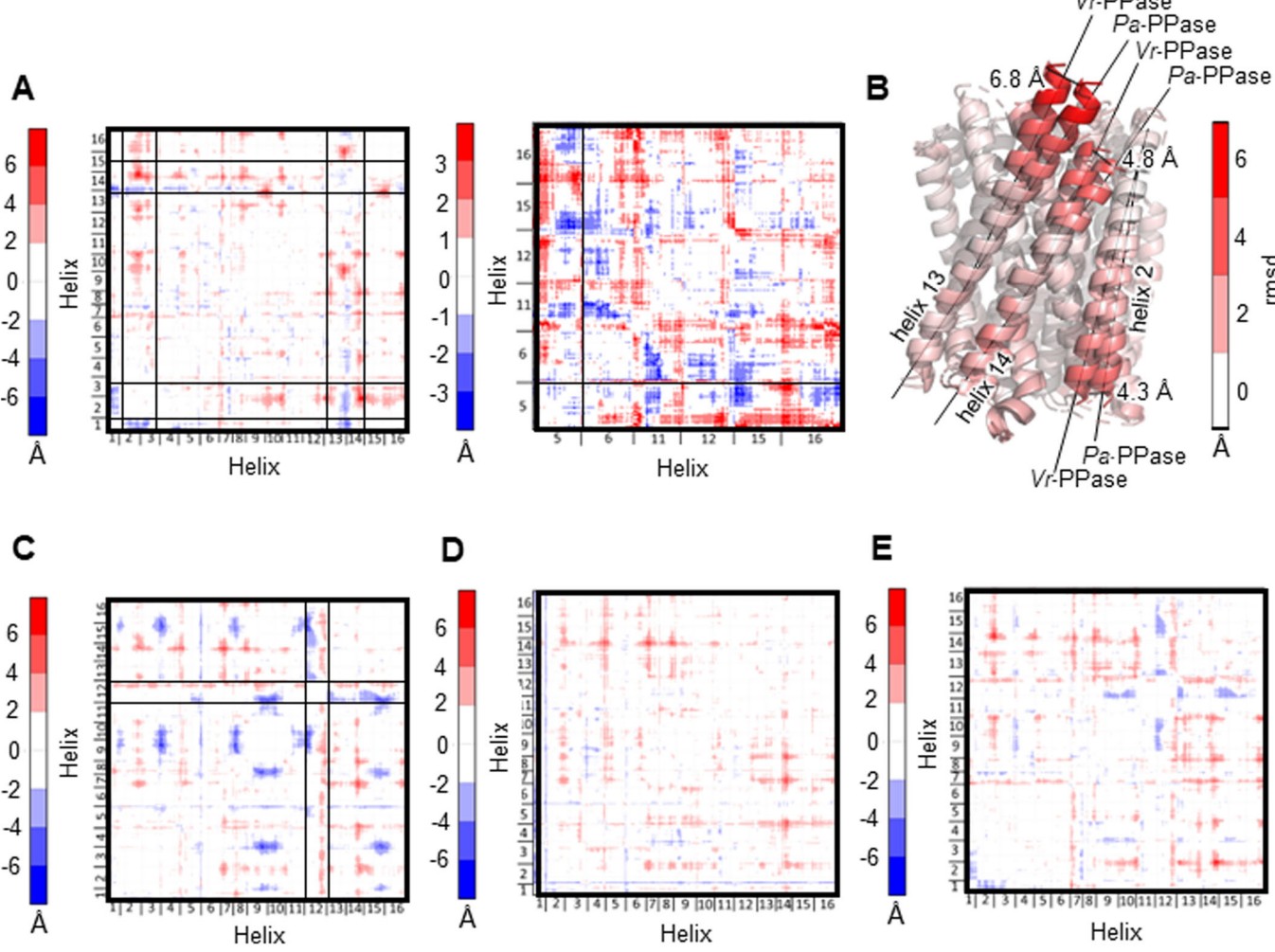

**Figure EV1. Comparison of inter-Cα distances between *Pa*PPase:Mg₅IDP and other M-PPase structures.**

The difference in inter-Cα distances is coloured from red (biggest difference) to blue (smallest difference) in each selection and helices with large clusters of changes are highlighted by black boxes. (A) Difference distance matrix (DiDiMa) of *Pa*PPase:Mg₅IDP (this study) versus *Vr*PPase:Mg₅IDP (PDB: 4A01). Left panel shows the DiDiMa of all atoms (scale: ±6 Å); right panel shows inter-atom differences of inner ring helices only (scale: ±3 Å). (B) Structural alignment of subunit A of the *Pa* and *Vr*PPase Mg₅IDP complexes, with helices coloured by their r.m.s.d./Cα. Dashed lines indicate the distances measured at the end of the helices. (C–E) DiDiMa (scale: ±6 Å) of *Pa*PPase:Mg₅IDP (this study) versus (C) *Tm*PPase:CaMg (PDB: 4AV3); (D) *Tm*PPase:Mg₄Pᵢ₂ (PDB: 4AV6) and (E) *Vr*PPase:Mg₂Pᵢ (PDB: 5GPJ).

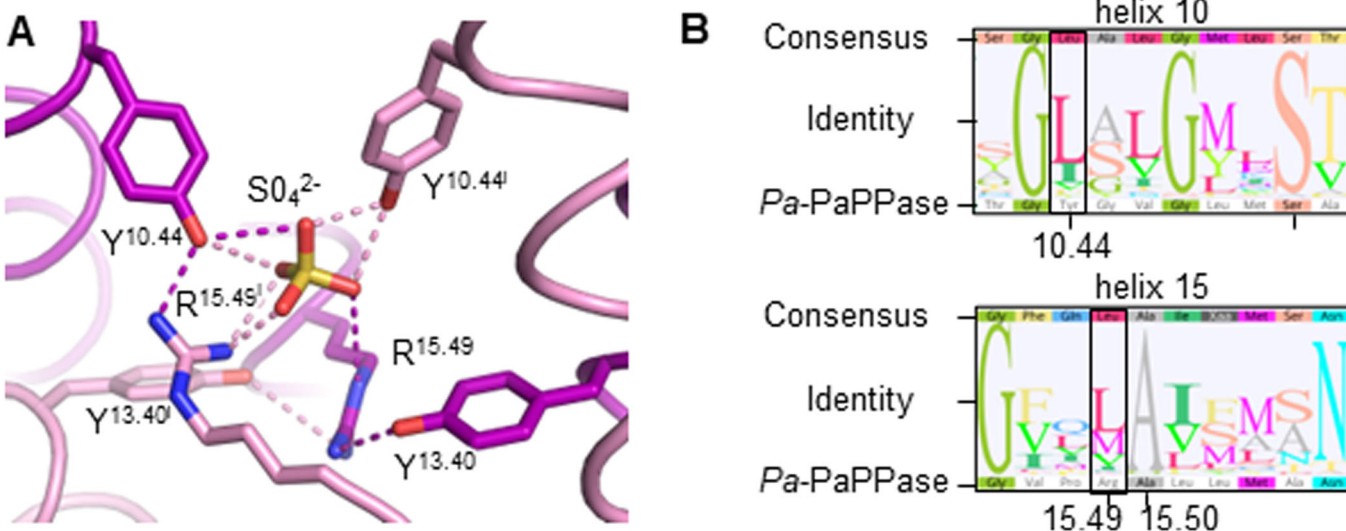

**Figure EV2.  SO₄²⁻ binding site at the dimer interface of *Pa*PPase:Mg₅IDP.**

(A) Structural overview with subunit A in purple and subunit B in pink (additionally marked with apostrophes). Side-chain interactions are shown as dashed lines. (B) Sequence analysis of the $SO_4^{2-}$ binding site. The consensus sequence and sequence identity (sequence logo showing the graphical representation of the residue conservation) are based on an alignment of 45 homologous sequences to *Pa*PPase identified in a blastp search of the UniProt database. Residues of interest are highlighted by a black box and labelled following the B&W convention.

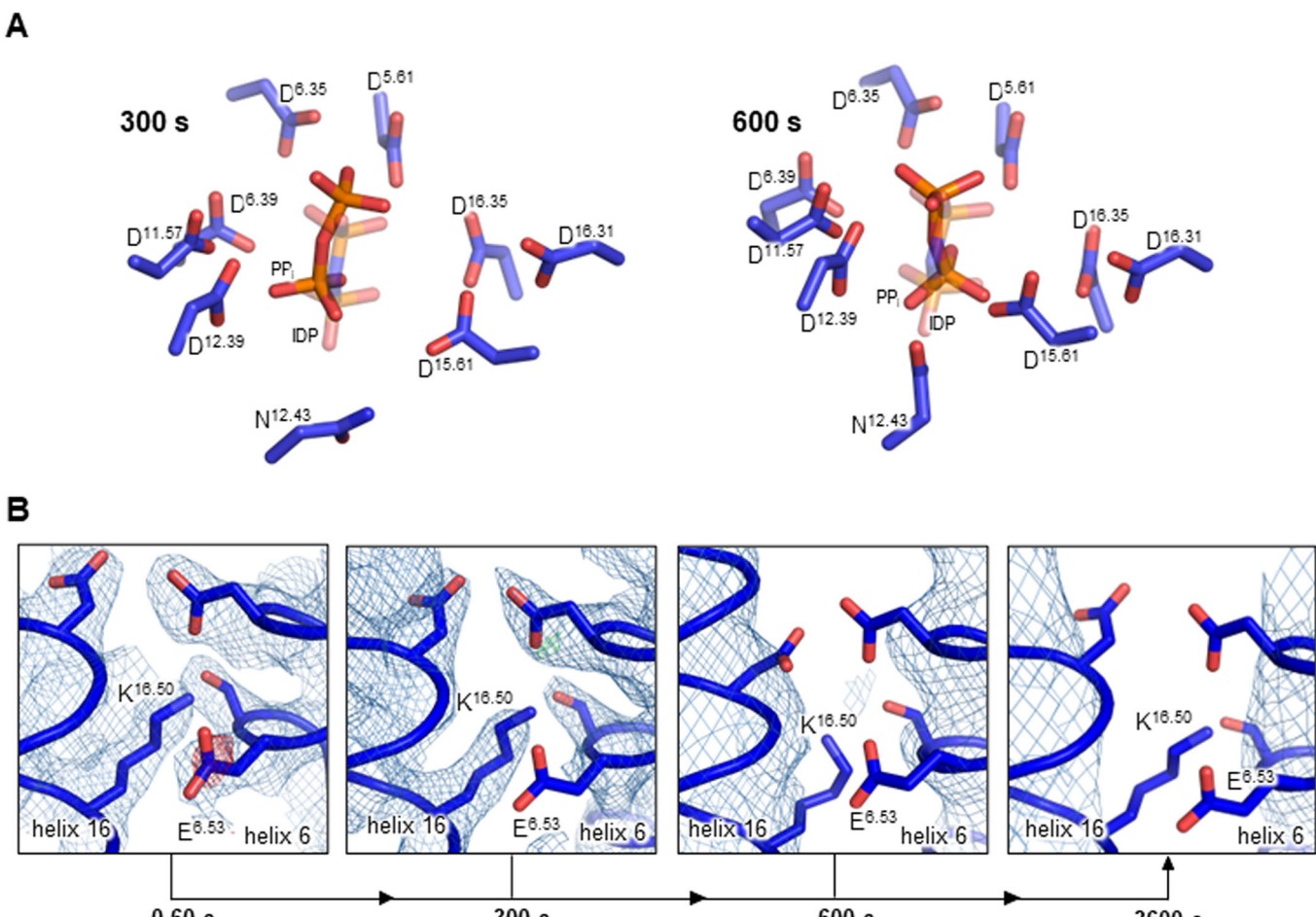

**Figure EV3. Key regions of time-resolved *Tm*PPase structures at different time points.**

(A) Comparison of ligand binding at the asymmetrically occupied active site in time-resolved structures after 300 s and 600 s reaction initiation (blue) and the symmetrically occupied active site in the static *Tm*PPase:Mg$_5$IDP structure (transparent, PDB: 5LZQ). The time-resolved structures have physiologically relevant PP$_i$ bound whereas the *Tm*PPase:Mg$_5$IDP structure has the non-hydrolysable substrate analogue IDP-bound. (B) Ion gate of time-resolved *Tm*PPase structures of grouped datasets of different time points (0–60 s) and combined datasets of the same time point (300, 600 and 3600 s). Structures show subunit A with 2mF$_o$-dF$_c$ density (blue) and mF$_o$-dF$_c$ density (red/green) at 1 σ and 3 σ, respectively. At low Na$^+$ concentrations (0–60 s), the semi-conserved glutamate appears disordered as indicated by negative mF$_o$-dF$_c$ density.

