## [Peer Review File · EMBO Reports]

Functional and structural asymmetry suggest a unifying principle for catalysis in membrane-bound pyrophosphatases

Jannik Strauss, Craig Wilkinson, Keni Vidilaseris, Orquidea De Castro Ribeiro, Jianing Liu, James Hillier, Maximilian Wichert, Anssi Malinen, Bernadette Gehl, Lars Jeuken, Arwen Pearson, and Adrian Goldman

DOI: [10.15252/embr.202358049](https://doi.org/10.15252/embr.202358049)

Corresponding author(s): Adrian Goldman (adrian.goldman@helsinki.fi), Jannik Strauss (jannik.strauss@mailbox.org), Adrian Goldman (adrian.goldman@helsinki.fi)

Review Timeline:

Transfer Date:	24th Aug 23
Editorial Decision:	25th Aug 23
Revision Received:	9th Oct 23
Editorial Decision:	24th Nov 23
Revision Received:	27th Nov 23
Accepted:	1st Dec 23

Transaction Report: This manuscript was transferred to EMBO reports following peer review at The EMBO Journal.

Referee #1:

The paper submitted to EMBL Journal entitled "Functional and structural asymmetry suggests a unifying principle for catalysis in membrane-bound pyrophosphatases"

The manuscript is nicely written with strong detail on the catalytic mechanism for the vectorial transport of sodium ions through the potassium-independent M-PPase. The authors claim that their findings explain both ion selectivity and suggest a plausible explanation for the reactivity displayed in only a subset of the ion sites.

- general summary and opinion about the principle significance of the study, its questions and findings

The authors present the first structure of a K⁺-independent M-PPase (from the thermophile *Pyrobaculum aerophilum*, determined at 3.1 Å resolution and at decreasing resolution ranges. The authors performed enzymatic assays on native PaPPase and three variants (A12.46K and A12.49T, and the double mutant), and conducted both electrometric and time-resolved crystallographic studies on TmPPase, a K⁺-dependent Na⁺-PPase.

- specific major concerns essential to be addressed to support the conclusions

While the study represents a strong case for a general mechanism for how the M-PPases function in general the diffraction data is severely hampered by anisotropy which hampers the interpretation of the structural data. This is particularly true for the completeness of the presented data at all the time slots deposited. Before moving further the datasets presented should be significantly improved and higher completeness collected. The Metrics for R_{free} and especially clash scores should be improved, this should still be possible at the resolution. The phase information carries a lot of information and phase bias might very well become an important issue here. If data like this is to be used with this quality additional dataset set should be collected which shows a similar trend using a longer wavelength to allow the detection of an anomalous signal arising from the presence of IDP.

- minor concerns that should be addressed

Write out the abbreviation IDP before it is used the first time

In the Figure 1 caption it states "IDP (imidodiphosphate)" This should be written as imidodiphosphate (IDP)" in the main text earlier in the manuscript.

Line 78 and 80, add ions after Mg²⁺

Line 81 A nucleophilic water molecule is poised to attack PPi -> should be a water molecule is positioned for nucleophilic attack on PPi

Line 180 "In previously solved IDP-bound", should be "In previously, determined IDP-bound"

Line 571 geometry (torsion, remove the parenthesis

Referee #2:

The authors reported the first crystal structure K⁺-independent membrane-bound pyrophosphatase, PaPPase, at the resolution of 3.8 angstrom. The authors performed a detailed structural analysis by comparing PaPPase and other solved PPase to answer the basis of ions specificity. The paper is well written, but I do have concerns about this manuscript:

1: I appreciate the careful thinking of model building for some key residues, for example, K 12.46. But as there is no side chain density at the resolution, I don't think the structure provides the structural detail of the potassium independence of Pappase. This situation also happened in the analysis of the ion selectivity part, assuming the conformation K16.50 is right and interacting with D6.50 and S6.54. But how are these residues related to proton pumping? Please specify it. In summary, I don't think the authors can get the solid structural information for ions specificity at this resolution.

2: To observe the asymmetry substrate binding of TmPPase, the authors designed time resolved crystallography. Indeed, the substrate occupies the activity site asymmetrically, finally symmetrically. What happened at the sodium binding sites in the two asymmetry subunit?

3: In the SSM assay, the authors used 100 micromolar of PPI and 50 micromolar IDP to activate the assay. Without information on the K_m of these substrates, how do the authors conclude: "TmPPase Pump two Na⁺ with PPI and 1 Na⁺ with IDP.

Other Minor points: in the figure panel 1B, can the author use another color for the bound Ca²⁺ in M3 or other sites? That will help the reader to distinguish Ca²⁺ and Mg²⁺ .

In figure 3: Can the author specify which part of the two models is used for alignment?

Referee #3:

The manuscript presents a crystal structure of a K⁺-independent membrane PPase, several crystal structures of a K⁺-dependent PPase collected at different timepoints during turnover and a limited number of biochemical and electrophysiological data. The combined data, together with previously published data are interpreted in the framework of a structural-mechanistic model. An important aspect of this mechanism is that asymmetry between the two protomers in the homodimeric complex is at the heart of the mechanism, with communication between the protomers essential, and pumping precedes hydrolysis of pyrophosphate.

-The amount of data presented is impressive, and in general I find the proposed mechanism plausible (although I am certainly not a familiar with the entire PPase field). I found the manuscript very difficult to read- probably because I am not an expert, and in places it seemed that the model - consistent as it may be internally- requires assumptions that are not directly deduced from the data. The authors should try and make the manuscript more accessible, and more specifically indicate where assumptions are made, and what kind of data will be needed to support the assumptions.

- line 87: "top half": replace with cytoplasmic half

- line 124/125: choose consistent nomenclature A12.46K or A/K 12.46

- I am concerned that the quality of the crystallographic data and resolution are not good enough to make some of the assignments: The authors should explicitly indicate for each water molecule and

Na ion that the build why other interpretations are less likely. It is not clear what their current assignments are based on.

- line 184 nomenclature: what is 10.33'

- line 208-209: it is very poor practice to build a model of a side-chain for which there is no density. Clearly the flexibility shows that there is no single conformation, so the model must reflect this, either by providing an ensemble, or by refraining from modelling. The interpretation of the K-independence is now entirely based on the model of a residue for which there is zero density. This must be removed.

- line 224-225: It is a giant leap to assume that the low affinity second binding site is indeed the binding site of a neighbouring protomer. This interpretation sounds over the top, at least in this place where it is based solely on the kinetic data.

- line 241-242: again modelling of a sidechain without density.

- line 359-360: I do not understand the reasoning. Hydrolysis is assumed to be slow, yet the protein pumps two ions when PP is added, and hydrolysis of 1 PP is needed to get both ions pumped. Is the rate of that first hydrolysis event fast? Or slow? Different from the second event?

Dear Adrian,

Thank you for your interest in transferring your research manuscript to our journal. As my colleague William Teale from The EMBO Journal had informed you on July 13, we are interested in the potential publication of your manuscript and asked you to revise it along the lines given below:

Please improve the structural data according to the suggestions from referee 1.

The conclusions on the structural details of K⁺ independence need to be toned down (ref 2, ref 3) and the building of side chains w/o density needs to be explained and justified in more detail (ref 2, ref 3). The conclusions derived from the electrometric measurements of ion pumping should be toned down appropriately (ref 2, ref 3) or alternatively further data, e.g. on Km should be provided. All minor concerns need to be addressed.

You have informed me that you have meanwhile addressed all concerns and are ready to upload a revised version. We have therefore transferred the manuscript to EMBO Reports on your behalf and I now send the official invitation letter so that you can upload the revised version to the EMBO Reports system.

We invite you to revise your manuscript with the understanding that the referee concerns (as detailed above and in their reports) must be addressed and their suggestions taken on board. Please address all referee concerns in a complete point-by-point response. Acceptance of the manuscript will depend on a positive outcome of a second round of review. It is EMBO Reports policy to allow a single round of revision only and acceptance or rejection of the manuscript will therefore depend on the completeness of your responses included in the next, final version of the manuscript.

You find all information on formatting requirements below. Since you are already to resubmit and I do not want to delay the process, I point out the most important things that we will need now:

- 1) The Data availability section must contain links that resolve directly to the dataset.
- 2) The figures must be uploaded as separate files, also the EV figures. Our system will build the merged PDF.

The rest looks fine at first glance and details can be sorted out at the next revision stage.

3) You will receive an e-mail from our data coordinator Hannah Sonntag, asking you to provide source data for your manuscript (point 8 below). In order not to delay the handling of your manuscript, please compile the source data requested in the checklist sent while your manuscript is under review. You can then upload it at the next revision stage.

All further details:

2) individual production quality figure files as .eps, .tif, .jpg (one file per figure).

Please download our Figure Preparation Guidelines (figure preparation pdf) from our Author Guidelines pages <https://www.embopress.org/page/journal/14693178/authorguide> for more info on how to prepare your figures.

4) a complete author checklist, which you can download from our author guidelines (<<https://www.embopress.org/page/journal/14693178/authorguide>>). Please insert information in the checklist that is also reflected in the manuscript. The completed author checklist will also be part of the RPF.

5) Please note that all corresponding authors are required to supply an ORCID ID for their name upon submission of a revised manuscript (<<https://orcid.org/>>). Please find instructions on how to link your ORCID ID to your account in our manuscript tracking system in our Author guidelines (<<https://www.embopress.org/page/journal/14693178/authorguide#authorshipguidelines>>)

6) We replaced Supplementary Information with Expanded View (EV) Figures and Tables that are collapsible/expandable online. A maximum of 5 EV Figures can be typeset. EV Figures should be cited as "Figure EV1, Figure EV2" etc... in the text and their respective legends should be included in the main text after the legends of regular figures.

7) Before submitting your revision, primary datasets (and computer code, where appropriate) produced in this study need to be deposited in an appropriate public database (see <<https://www.embopress.org/page/journal/14693178/authorguide#dataavailability>>).

The accession numbers and database should be listed in a formal "Data Availability" section (placed after Materials & Method) that follows the model below (see also <<https://www.embopress.org/page/journal/14693178/authorguide#dataavailability>>). Please note that the Data Availability Section is restricted to new primary data that are part of this study.

Data availability

Additional information on source data and instruction on how to label the files are available <<https://www.embopress.org/page/journal/14693178/authorguide#sourcedata>>.

10) Figure legends and data quantification:

- the name of the statistical test used to generate error bars and P values,
- the number (n) of independent experiments (please specify technical or biological replicates) underlying each data point,
- the nature of the bars and error bars (s.d., s.e.m.)
- If the data are obtained from n {less than or equal to} 5, show the individual data points in addition to the SD or SEM.
- If the data are obtained from n {less than or equal to} 2, use scatter blots showing the individual data points.

11) Our journal encourages inclusion of *data citations in the reference list* to directly cite datasets that were re-used and obtained from public databases. Data citations in the article text are distinct from normal bibliographical citations and should directly link to the database records from which the data can be accessed. In the main text, data citations are formatted as follows: "Data ref: Smith et al, 2001" or "Data ref: NCBI Sequence Read Archive PRJNA342805, 2017". In the Reference list, data citations must be labeled with "[DATASET]". A data reference must provide the database name, accession

number/identifiers and a resolvable link to the landing page from which the data can be accessed at the end of the reference. Further instructions are available at <<https://www.embopress.org/page/journal/14693178/authorguide#referencesformat>>.

12) As part of the EMBO publication's Transparent Editorial Process, EMBO Reports publishes online a Review Process File to accompany accepted manuscripts. This File will be published in conjunction with your paper and will include the referee reports, your point-by-point response and all pertinent correspondence relating to the manuscript.

Kind regards,

Martina

Point-by-point response letter EMBO R submission: EMBOR-2023-58049-T

We would like to thank the referees for their thoughtful evaluation of our work, and we are confident that the revised manuscript adequately addresses the concerns and suggestions put forth during the review process. We believe that the revision process significantly improved the quality and rigor of the study, and we are hopeful that it will be deemed suitable for publication in EMBO Reports.

Thank you for your continued consideration of our manuscript at EMBO Reports. We eagerly await your feedback on the revisions, and we remain open to further minor modifications if deemed necessary.

Referee #1:

The manuscript is nicely written with strong detail on the catalytic mechanism for the vectorial transport of sodium ions through the potassium-independent M-PPase. The authors claim that their findings explain both ion selectivity and suggest a plausible explanation for the reactivity displayed in only a subset of the ion sites.

The authors present the first structure of a K⁺-independent M-PPase (from the thermophile *Pyrobaculum aerophilum*, determined at 3.1 Å resolution and at decreasing resolution ranges. The authors performed enzymatic assays on native PaPPase and three variants (A12.46K and A12.49T, and the double mutant), and conducted both electrometric and time-resolved crystallographic studies on TmPPase, a K⁺-dependent Na⁺-PPase.

Comment 1: While the study represents a strong case for a general mechanism for how the M-PPases function in general the diffraction data is severely hampered by anisotropy which hampers the interpretation of the structural data. This is particularly true for the completeness of the presented data at all the time slots deposited. Before moving further the datasets presented should be significantly improved and higher completeness collected. The Metrics for R_{free} and especially clash scores should be improved, this should still be possible at the resolution. The phase information carries a lot of information and phase bias might very well become an important issue here. If data like this is to be used with this quality additional dataset set should be collected which shows a similar trend using a longer wavelength to allow the detection of an anomalous signal arising from the presence of IDP.

The data sets presented here represent the very best data that were able to be collected out of hundreds of tries; in case of PaPPase over the course of >8 years. In our opinion, the reviewer is basing his opinions about what can be achieved on simpler - possibly non-membrane - proteins. Membrane protein crystals are quite often highly anisotropic (Robert et al. Sci Rep 7, 17013 (2017), <https://doi.org/10.1038/s41598-017-17216-1>) and so the kinds of problems that remain in the structure are not easily, or at all, solvable. Moreover, we would like to stress that the crystallographic data of TmPPase, which was flagged by the referee, are one of the very few examples of *time-resolved* structural studies of membrane proteins. Given the extremely challenging nature of these kind of experiments, if not working with robust and simple light-activated pumps like bacteriorhodopsin, we disagree with the referee's assessment of the TmPPase metrics. With the exception of the 600s TmPPase structure, all of the metrics are at about the 40th percentile or better *for* structures at this resolution (0% the worst; 100% the best). It is the nature of distributions that some structures are worse than others, and being less than 1 σ worse than the mean in these kind of experiments is in our opinion OK.

Nevertheless, we went back to refinement using BUSTER instead of Phenix, a refinement software that is reported to cope well with low-resolution structures, in order to further improve our models according to the referees suggestions. We also tried using PDBredo. However, both approaches were not successful as neither the electron density nor the refinement metrics improved. Collecting better datasets is anything

but trivial for membrane proteins, especially for time-resolved studies, and in our opinion out of scope for a revision process.

Although we disagree with the referee's assessment of the *Tm*PPase metrics given the extremely challenging nature of time-resolved studies with membrane proteins, we are grateful for pointing out their worries as future readers may have similar concerns. We therefore added a paragraph that specifically comments on the diffraction data quality and clarifies on which data our proposed model of M-PPase catalysis is based on to make clear that it does not require very high-resolution data and is further backed up by complementary functional studies. The paragraph reads:

"We are aware that higher-resolution data would enable a more in-depth analysis of the mechanistic details of pumping, excess substrate inhibition and inter-subunit communication. Extensive efforts were dedicated to improving the resolution and hundreds of crystals were tested. Only a handful of crystals were obtained that diffracted sufficiently to be combined and yield the time-resolved structures presented in this study. At the obtained resolution, the structures still give a reliable snap shot of the global enzyme conformation at different time points and a number of clear mechanistic details are observed. The 300- and 600-seconds structure clearly indicate asymmetric occupation of the active site for the first time, while the 3600-seconds data demonstrates unrestricted access to the active site to prove that that asymmetric substrate binding is not simply a crystallographic artefact.

These unambiguous observations are consistent with the half-of-the-sites reactivity seen in kinetic assays (Anashkin et al., 2021; Artukka et al., 2018; Vidilaseris et al., 2019); they provide snapshots of a structural binding pathway, and support a new comprehensive kinetic model of catalysis (see Discussion)."

Comment 2: Write out the abbreviation IDP before it is used the first time. In the Figure 1 caption it states "IDP (imidodiphosphate)" This should be written as imidodiphosphate (IDP)" in the main text earlier in the manuscript.

IDP was written out where first mentioned in the text as suggested.

Comment 3: Line 78 and 80, add ions after Mg²⁺

We have added the word ions after Mg²⁺ as suggested.

Comment 4: Line 81 A nucleophilic water molecule is poised to attack P_{Pi} -> should be a water molecule is positioned for nucleophilic attack on P_{Pi}

The sentence was changed according to the reviewers suggestion.

Comment 5: Line 180 "In previously solved IDP-bound", should be "In previously, determined IDP-bound"

The sentence was change to improve clarity and grammar.

Comment 6: Line 571 geometry (torsion, remove the parenthesis

We have removed the parenthesis as suggested.

Referee #2:

The authors reported the first crystal structure K⁺-independent membrane-bound pyrophosphatase, PaPPase, at the resolution of 3.8 angstrom. The authors performed a detailed structural analysis by comparing PaPPase and other solved PPase to answer the basis of ions specificity. The paper is well written, but I do have concerns about this manuscript:

Comment 1: 1: I appreciate the careful thinking of model building for some key residues, for example, K^{12.46}. But as there is no side chain density at the resolution, I don't think the structure provides the structural detail of the potassium independence of Pappase. This situation also happened in the analysis of the ion selectivity part, assuming the conformation K16.50 is right and interacting with D6.50 and S6.54. But how are these residues related to proton pumping? Please specify it. In summary, I don't think the authors can get the solid structural information for ions specificity at this resolution.

We would like to thank the referee for their in-depth review of our structural and functional data. Following their feedback, we have carefully re-inspected the active site and ion gate region of the PaPPase structure and internally discussed the concerns raised. We agree that in the absence of electron density for K^{12.46}, there is no clear evidence for K⁺-independence, even if it could be explained by modelling the side chain in the stereochemically most plausible pose. We therefore decided to tone down the respective sections and further elaborated on how we handled modelling of side chain residues at key catalytic regions in absence of sufficient electron density. It now reads:

“We were not able to confirm definitively the kinetic data that place K^{12.46} at the heart of K⁺-independence structurally, as residues with flexible side chains such as lysines lacked sufficient electron density at the given resolution (Appendix Fig S3). There is an ongoing debate among crystallographers about how to handle disordered sidechains in model building (Lamb et al., 2015). One can simply remove side chains for which there is no experimental evidence, *i.e.* electron density, to avoid bias and overinterpretation of data; however, the truncation of side chains may lead to confusion about amino acid identity for non-expert viewers, does not reflect the real chemical entity (Lamb et al., 2015) and may lead to other errors in interpretation – the encroachment of other residues into the region with missing atoms. It is reasonable to argue that the sidechains should be included in a refined structure as they are most definitely chemically present and that the resolved atoms both locally and globally constrain the orientation of individual side chain atoms. Following this rationale, the side chain is placed in the stereochemically most plausible pose, which gives a good approximation of at least one potential real position, especially if residues adjacent in three dimensions are well defined. This approach avoids confusion about amino acid identity, but obliges the viewer to check electron density maps and B-factors, which will refine to high values in absence of electron density, to determine the model reliability in the region of interest (Lamb et al., 2015). We therefore decided to model poorly defined side chains in catalytic key regions in their stereochemically most plausible pose (Appendix Table S5-6) and allow their refinement to high B-factors, as they are chemically present in the enzyme, but explicitly flag the absence of sufficient density in the text where appropriate and do not base our mechanistic models on such side chain orientations. For K^{12.46}, this indeed places ϵ -NH₃⁺ of K^{12.46} at the K⁺-binding site (Figure 4B), which would explain K⁺-independence. It is worth noting (that the most likely poses are clearly energetically separated even from the second most likely poses in terms of clash score and hydrogen bonds (Appendix Table S5). Nevertheless, a more robust analysis requires higher resolution data.”

We have also updated the legend of Figure 4B to emphasize again that K^{12.46} is modelled based on the stereochemical most plausible pose. It now reads:

“K⁺/K^{12.46} cationic centre with K⁺ (transparent purple sphere) and nucleophilic water (transparent red sphere) modelled into the structure based on its position in

VrPPase:Mg₅IDP (PDB: 4A01). K^{12.46} is not defined in the electron density but modelled in the stereochemically most plausible pose (Table S5), which would allow it to substitute for K⁺.”

We hope that our toned down data interpretation, the detailed reasoning for modelling the stereochemically most plausible side chain pose and the explicit warnings about missing electron density satisfy the referee’s concerns.

As for the model ion-selectivity, we also agree with the referee that it cannot be based on K^{16.50} *alone* for the same reasons. Although K^{16.50} is certainly part of the ion gate, our model of ion selectivity does not rely solely on its side chain orientation, but rather depends on the interplay of the semi-conserved glutamate E^{6.53/57} with S^{5.43}, which is governed by helix 5 orientation, and all other residues (D^{6.50}, S^{6.54}, N^{16.46}) forming the ion binding site. All of these residues are defined in the electron density at 1 σ and back up our model of ion-selectivity. We now see that this was not clear before and have therefore adjusted the respective manuscript section. It now reads:

“The structure of the ion gate must hold the explanation to ion selectivity, but the current model, that the position of the semi-conserved glutamate alone defines ion selectivity does not hold for K⁺-independent H⁺-PPases (see Introduction). Our *Pa*PPase:Mg₅IDP structure provides the first data on the ion gate set up of K⁺-independent H⁺-PPases with E^{6.53} and shows that the 3D orientation of ion gate residues D^{6.50}, S^{6.54}, N^{16.46} and E^{6.53} in particular *all* resemble the structure of VrPPase:Mg₅IDP, a H⁺-pump with E^{6.57}, and not *Tm*PPase:Mg₅IDP, a Na⁺-pump with E^{6.53}, despite the shift of the semi-conserved glutamate (Figure 5A). This raises the question: What causes E^{6.53} to reorient in *Pa*PPase so that it forms the same interactions as E^{6.57} in VrPPase, not as E^{6.53} in *Tm*PPase (Figure 5B)?

In *Pa*PPase:Mg₅IDP, helix 5 is straightened (see above) and moved out of the protein core at the ion gate by about 2 Å compared with other IDP-bound M-PPases (Figure 5C-D). This allows S^{5.43} to hydrogen bond to E^{6.53}, as occurs with E^{6.57} in VrPPase:Mg₅IDP (Figure 1D). In *Tm*PPase, helix 5 is closer to helix 6 and helix 16, forcing position 6.53 to point away from S^{5.43}, thereby contributing to the formation of a Na⁺-binding site (*Tm*PPase:Mg₅IDP) (Figure 5C).”

As part of the ion gate, the orientation of K^{16.50} is indeed of interest and thus also depicted in Figure 5b, but it does not dictate ion selectivity in the activated enzyme state. To make sure that the reader is aware of the missing density for K^{16.50} when inspecting Figure 5D, the respective figure legend now reads:

“Close-up view and comparison of the semi-conserved glutamate (E^{6.53/57}) orientation and helix 5 conformation in *Pa*PPase:Mg₅IDP and VrPPase:Mg₅IDP. Dashed lines shown the coordination of key residues involved in ion selectivity. K^{16.50} is not defined in the electron density but modelled in the stereochemically most plausible pose (Table S6), which aligns with its orientation in VrPPase:Mg₅IDP. Major structural changes are indicated by black arrows”

Comment 2: To observe the asymmetry substrate binding of TmPPase, the authors designed time resolved crystallography. Indeed, the substrate occupies the activity site asymmetrically, finally symmetrically. What happened at the sodium binding sites in the two asymmetry subunit.

The observed changes in subunit A are typically linked to the localisation of a Na⁺ ion at the ion gate (Li et al, Nature Communications, 7, 1–11, 2016); however, due to the extremely challenging nature of time-resolved X-ray crystallographic studies with membrane proteins, we were not able to collect datasets at high enough resolution to resolve Na⁺ ions, which are isoelectronic with water molecules. Nevertheless, upon binding of substrate in its correct binding pose (t=600 seconds), subunit A transitioned from the resting state in the active state as described in the manuscript.

Comment 3: In the SSM assay, the authors used 100 micromolar of PP_i and 50 micromolar IDP to activate the assay. Without information on the K_m of these substrates, how do the authors conclude: "TmPPase Pump two Na⁺ with PP_i and 1 Na⁺ with IDP.

IDP is not a substrate in mPPases, but a competitive inhibitor so a K_m cannot be determined. A full catalytic cycle, which would in all models involve pumping from both active sites, *i.e.* 2 Na⁺, can therefore not be achieved with IDP as it is not hydrolysed, a prerequisite for getting the second subunit ready for pumping. In other words, ion and substrate binding to subunit B cannot take place until hydrolysis happens in subunit A – and no hydrolysis can happen with IDP. The key fact is that PP_i leads to the expected pumping activity, IDP leads to some pumping activity, but the very close isostere etidronate does not, indicating a significant difference in their behaviour. We showed this for *Vigna radiata* (mung bean) mPPase in 2016 (Li et al., *Nat Commun* 2016; 7: 13596). This is explained in the following sentences:

“The measured currents in presence of IDP are specifically caused by a single ion pumping event and not by Mg²⁺ being brought slightly beneath the membrane surface as PP_i mimics that allow loop₅₋₆ closure and thus pumping (*e.g.* IDP) induce a bigger signal than those that do not (*e.g.* etidronate), and these signals can be collapsed by H⁺-specific ionophores (Li et al., 2016, Shah et al., 2017).”

Nevertheless, we would like to thank the reviewer for pointing out that the use of 100 micromolar PP_i, but 50 micromolar IDP weakens our data interpretation. To ensure that the differences in SSM results are not due to a lower affinity of IDP for TmPPase, we have repeated the experiment with 100 micromolar IDP and updated the respective figure panel and methods section. We observed no significant difference to the previously collected data, which unambiguously shows that at ≥50 micromolar IDP binding is already saturated and that our interpretation of the data holds. We believe that the result of the additional experiment thus satisfies the referees concern.

Comment 4: Other Minor points: in the figure panel 1B, can the author use another color for the bound Ca²⁺ in M3 or other sites? That will help the reader to distinguish Ca²⁺ and Mg²⁺.

We have adjusted the colour of the Ca²⁺ ion as suggested by the reviewer.

Comment 5: In figure 3: Can the author specify which part of the two models is used for alignment?

The alignment is based on Cα atoms of subunit A of the respective structures. We have updated the figure legend to make this clear.

Referee #3:

The manuscript presents a crystal structure of a K⁺-independent membrane PPase, several crystal structures of a K⁺-dependent PPase collected at different timepoints during turnover and a limited number of biochemical and electrophysiological data. The combined data, together with previously published data are interpreted in the framework of a structural-mechanistic model. An important aspect of this mechanism is that asymmetry between the two protomers in the homodimeric complex is at the heart of the mechanism, with communication between the protomers essential, and pumping precedes hydrolysis of pyrophosphate.

Comment 1: The amount of data presented is impressive, and in general I find the proposed mechanism plausible (although I am certainly not a familiar with the entire PPase field). I found the manuscript very difficult to read- probably because I am not an expert, and in places it seemed that the model -consistent as it may be internally- requires assumptions that are not directly deduced from the data. The authors should try and make the manuscript more accessible, and more specifically indicate where assumptions are made, and what kind of data will be needed to support the assumptions.

We would like to thank the referee for acknowledging the large amount of data presented in this study. We are aware that for non-expert readers, the amount of information provided is difficult to digest. However, the manuscript must cover the current mechanistic models of several aspects of M-PPase biology, including half-of-the-sites-reactivity, ion-selectivity and energy-coupling to provide all background information necessary for evaluating our proposed *unified* model of M-PPase catalysis. Similarly, further simplification of the text would mean leaving out important details that the M-PPase community needs to independently assess the validity of our claims. Having the referees request in mind, we have tried our best to clarify our claims, remove less-relevant and redundant information, remove claims that are based on assumptions and indicate where more data is required to support our interpretation. The changes to the respective manuscript sections are listed below (A-I), if not addressed in our response to the referee's comments 2-9.

A) In the "Half-of-the-sites-reactivity" section of the discussion we have removed following sentences as well as the associated Figure (Appendix Fig S6) for simplicity as we deemed the provided information not critical for our proposed model:

"Intriguingly, the length of the exit channel loops appears to be conserved: If exit channel loop8-9 is long (12 18 residues), loop10-11 is short (3 9 residues) and vice versa (Appendix Fig S6)."

B) We have added a sentence to the last paragraph of the "Half-of-the-sites-reactivity" section of the discussion to clarify the need for higher resolution data to back up a active site occupation of PP_i in subunit A and $(P_i)_2$ in subunit B.

"In the context of a half-of-the-sites reactivity mechanism, we propose that this represents a configuration in which PP_i is bound in one subunit and $(P_i)_2$ in the other (Figure 9). However, the resolution of the structure is only 4.53 Å: higher resolution time-resolved structural data is required to confirm this."

C) We would like to point out that our interpretation of the time-resolved crystallographic data does not rely on mapping individual side chain orientations in high-resolution structures. We therefore added a paragraph to the results section of "Direct observation of asymmetric PP_i binding in *Tm*PPase" that specifically comments on the diffraction data quality and clarifies on which structural data (alongside the provided functional data) our proposed model of M-PPase catalysis is based on. The paragraph reads:

"We are aware that higher-resolution data would enable a more in-depth analysis of the mechanistic details of pumping, excess substrate inhibition and inter-subunit communication. Extensive efforts were dedicated to improving the resolution and hundreds of crystals were tested. Only a handful of crystals were obtained that diffracted sufficiently to be combined and yield the time-resolved structures presented in this study. At the obtained resolution, the structures still give a reliable snap shot of the global enzyme conformation at different time points and a number of clear mechanistic details are observed. The 300- and 600-seconds structure clearly indicate asymmetric occupation of the active site for the first time, while the 3600-seconds data demonstrates unrestricted access to the active site to prove that that asymmetric substrate binding is not simply a crystallographic artefact."

These unambiguous observations are consistent with the half-of-the-sites reactivity seen in kinetic assays (Anashkin et al., 2021; Artukka et al., 2018; Vidilaseris et al., 2019); they provide snapshots of a structural binding pathway, and support a new comprehensive kinetic model of catalysis (see Discussion).”

D) We have toned down our claims about presenting first structural evidence for K^+ -independence due to the lack of sufficient electron density for $K^{12.46}$ (see comment 6 + response) as requested and elaborated on our reasoning for modelling the side chains of residues $K^{12.46}$ & $K^{16.50}$.

E) We have also tried to make clear that the proposed mechanism for ion selectivity does not rely on assumptions, but is backed up by the structural data provided. Moreover, we have rewritten the respective section to make it easier to understand (see comment 8 + response).

F) We have removed following sentence from the “Ion selectivity” discussion section as it did not add much valuable information, but could be confusing to readers outside the M-PPase field:

“In all these structures, ion binding in subunit B would not occur before pumping and hydrolysis in subunit A, but the structure of the ion binding site would be the same.”

G) We have also deleted following sentence about the ion gate configuration in Na^+ -pumps at low Na^+ concentrations from the “Ion selectivity” discussion section as it was somewhat redundant:

“This may also be possible in K^+ -dependent Na^+ -PPases at low Na^+ concentrations ($E^{6.53}$ disordered in 0-60 s structure) as suggested above.”

H) We have added following sentence stating the experiments needed to prove one of our assumptions about asymmetrical ion gate orientations in the “ion selectivity” of the discussion. It now reads:

“To confirm this, the complete catalytic cycle must be mapped in time resolved structural studies at high enough resolution in to resolve individual side chain orientations”

I) We have tried to simplify our explanation of the data of electrometric measurements (see comment 9+response).

We think that the changes made to the manuscript based on the comments of the referee have made it more accessible to non-expert readers and clarify on what data our proposed model of catalysis is based on. We hope that our changes satisfy the core of the referees request.

Comment 2: - line 87: "top half": replace with cytoplasmic half

We have now implemented the changes suggested by the reviewer.

Comment 3: - line 124/125: choose consistent nomenclature A12.46K or A/K 12.46

A12.46K refers to the mutation of alanine to lysine at position 12.46, whereas A/K12.46 refers to the position 12.46, which may alanine or lysine depending on the M-PPase. To be more clear we have changed the sentence which now reads” Of them, the change at position 12.46 is the one that defines K^+ -dependence “.

Comment 4: I am concerned that the quality of the crystallographic data and resolution are not good enough to make some of the assignments: The authors should explicitly indicate for each water molecule and Na ion that the build why other interpretations are less likely. In is not clear what their current assignments are based on.

The referee is correct: the resolution is insufficient to build Na^+ ions as opposed to water molecules – and therefore none were built. We can only hypothesis if the reviewer meant to refer to Mg^{2+} ions, which were build as Mg-IDP complex into the electron density at the highly conserved active site in alignment with published functional data, the available high-resolution reference structures *Tm*PPase:Mg₅IDP and

VrPPase:Mg₅IDP and the expected coordination geometry. There is no reason to assume a fundamentally different binding of Mg₅IDP in our new structures as compared to the reference structures. Besides Mg²⁺ ions, we only built water molecules in places where there was reasonable electron density. Of these, the PaPPase structure is the only one with a few structural water molecules placed at the ion gate and active site. These were built with an acceptable electron density fit and coordination geometry and in alignment with the local B-factor distribution and high-resolution reference structures in the same enzyme state, e.g. VrPPase:Mg₅IDP. We did not comment on the placement of structural (or solvent) water molecules as they do not play a role in our proposed model of catalysis.

Comment 5: - line 184 nomenclature: what is 10.33'

We have made this sentence more clear, which now reads:

“This enables propagation of motions from the inner to the outer ring and into the other subunit (indicated by apostrophe) *via* E^{5.71}-R^{13.62}-R/I/K^{10.33}”

Comment 6: - line 208-209: it is very poor practice to build a model of a side-chain for which there is no density. Clearly the flexibility shows that there is no single conformation, so the model must reflect this, either by providing an ensemble, or by refraining from modelling. The interpretation of the K-independence is now entirely based on the model of a residue for which there is zero density. This must be removed.

We would like to thank the referee for their thorough review of our structural data. Following their feedback, we have carefully re-inspected the active site and ion gate region of the PaPPase structure and internally discussed the concerns raised. We agree that in the absence of electron density for K^{12.46}, there is no clear evidence for K⁺-independence, even if it could be explained by modelling the side chain in the stereochemically most plausible pose. We therefore toned down the respective sections and further elaborated on how we handled modelling of side chain residues at key catalytic regions in absence of sufficient electron density. It now reads:

“We were not able to confirm definitively the kinetic data that place K^{12.46} at the heart of K⁺-independence structurally, as residues with flexible side chains such as lysines lacked sufficient electron density at the given resolution (Appendix Fig S3). There is an ongoing debate among crystallographers about how to handle disordered sidechains in model building (Lamb et al., 2015). One can simply remove side chains for which there is no experimental evidence, *i.e.* electron density, to avoid bias and overinterpretation of data; however, the truncation of side chains may lead to confusion about amino acid identity for non-expert viewers, does not reflect the real chemical entity (Lamb et al., 2015) and may lead to other errors in interpretation – the encroachment of other residues into the region with missing atoms. It is reasonable to argue that the sidechains should be included in a refined structure as they are most definitely chemically present and that the resolved atoms both locally and globally constrain the orientation of individual side chain atoms. Following this rationale, the side chain is placed in the stereochemically most plausible pose, which gives a good approximation of at least one potential real position, especially if residues adjacent in three dimensions are well defined. This approach avoids confusion about amino acid identity, but obliges the viewer to check electron density maps and B-factors, which will refine to high values in absence of electron density, to determine the model reliability in the region of interest (Lamb et al., 2015). We therefore decided to model poorly defined side chains in catalytic key regions in their stereochemically most plausible pose (Appendix Table S5-6) and allow their refinement to high B-factors, as they are chemically present in the enzyme, but explicitly flag the

absence of sufficient density in the text where appropriate and do not base our mechanistic models on such side chain orientations. For $K^{12.46}$, this indeed places $\epsilon\text{-NH}_3^+$ of $K^{12.46}$ at the K^+ -binding site (Figure 4B), which would explain K^+ -independence. It is worth noting (that the most likely poses are clearly energetically separated even from the second most likely poses in terms of clash score and hydrogen bonds (Appendix Table S5). Nevertheless, a more robust analysis requires higher resolution data.”

We have also updated the legend of Figure 4B to emphasize again that $K^{12.46}$ is modelled based on the stereochemical most plausible pose. It now reads:

“ $K^+/K^{12.46}$ cationic centre with K^+ (transparent purple sphere) and nucleophilic water (transparent red sphere) modelled into the structure based on its position in VrPPase:Mg₅IDP (PDB: 4A01). $K^{12.46}$ is not defined in the electron density but modelled in the stereochemically most plausible pose (Appendix Table S5), which would allow it to substitute for K^+ .”

We hope that our toned down data interpretation, the detailed reasoning for modelling the stereochemically most plausible side chain pose and the explicit warnings about missing electron density satisfy the referee’s concerns.

Comment 7: line 224-225: It is a giant leap to assume that the low affinity second binding site is indeed the binding site of a neighbouring protomer. This interpretation sounds over the top, at least in this place where it is based solely on the kinetic data.

We respectfully disagree with the referee’s interpretation. The kinetic finding of half-of-the-sites reactivity is solid and broadly accepted by M-PPase experts, as shown in our work and in several papers (Artukka et al. (2018) *Biochemical Journal*, 475(6), 1141–1158. <https://doi.org/10.1042/BCJ20180071>, Vidilaseris et al. (2019). *Science Advances*, 5(5). <https://doi.org/10.1126/sciadv.aav7574>, Anashkin et al. (2021). *International Journal of Molecular Sciences*, 22, 9820. <https://doi.org/10.3390/ijms22189820>). Such a kinetic finding does, indeed, mean that binding of the second substrate is either weaker or has a lower k_{cat} than the first and it would be quixotic to imagine that the binding event (that can lead to hydrolysis: V2 in Table 2 is not zero) is somewhere other than the active site. The exact nature of substrate inhibition is still unclear, and appears to be mutation- and enzyme-dependent as shown in our data and in the data from Anashkin et al. (2021). *International Journal of Molecular Sciences*, 22, 9820. <https://doi.org/10.3390/ijms22189820>. We have updated the respective manuscript section and added a sentence plus references to clarify this:

“The nature of the kinetic scheme means that the second binding site reported for PP_i must be in the other monomer, as it can be hydrolysed (it corresponds to the E₂S₂ complex). Half-of-the-sites reactivity has been reported before on a variety of M-PPase (Artukka et al., 2018; Vidilaseris et al., 2019; Anashkin et al., 2021).”

Comment 8: - line 241-242: again modelling of a sidechain without density.

As mentioned above, there is an ongoing debate about how to handle disordered sidechains in model building among crystallographers and each method has its advantages and disadvantages. We have elaborated on our reasoning for modelling missing side chain densities but agree with the referee that it cannot be based on $K^{16.50}$ alone for the same reasons. Although $K^{16.50}$ is certainly part of the ion gate, our model of ion selectivity does not rely solely on its side chain orientation, but rather depends on the

interplay of the semi-conserved glutamate E^{6.53/57} with S^{5.43}, which is governed by helix 5 orientation, and all other residues (D^{6.50}, S^{6.54}, N^{16.46}) forming the ion binding site. All of these residues are defined in the electron density at 1 σ and back up our model of ion-selectivity. We now see that this was not clear before and have therefore adjusted the respective manuscript section. It now reads:

“The structure of the ion gate must hold the explanation to ion selectivity, but the current model, that the position of the semi-conserved glutamate alone defines ion selectivity does not hold for K⁺-independent H⁺-PPases (see Introduction). Our *Pa*PPase:Mg₅IDP structure provides the first data on the ion gate set up of K⁺-independent H⁺-PPases with E^{6.53} and shows that the 3D orientation of ion gate residues D^{6.50}, S^{6.54}, N^{16.46} and E^{6.53} in particular *all* resemble the structure of *Vr*PPase:Mg₅IDP, a H⁺-pump with E^{6.57}, and not *Tm*PPase:Mg₅IDP, a Na⁺-pump with E^{6.53}, despite the shift of the semi-conserved glutamate (Figure 5A). This raises the question: What causes E^{6.53} to reorient in *Pa*PPase so that it forms the same interactions as E^{6.57} in *Vr*PPase, not as E^{6.53} in *Tm*PPase (Figure 5B)?

In *Pa*PPase:Mg₅IDP, helix 5 is straightened (see above) and moved out of the protein core at the ion gate by about 2 Å compared with other IDP-bound M-PPases (Figure 5C-D). This allows S^{5.43} to hydrogen bond to E^{6.53}, as occurs with E^{6.57} in *Vr*PPase:Mg₅IDP (Figure 1D). In *Tm*PPase, helix 5 is closer to helix 6 and helix 16, forcing position 6.53 to point away from S^{5.43}, thereby contributing to the formation of a Na⁺-binding site (*Tm*PPase:Mg₅IDP) (Figure 5C).”

As part of the ion gate, the orientation of K^{16.50} is indeed of interest and thus also depicted in Figure 5b, but it does not dictate ion selectivity in the activated enzyme state. To make sure that the reader is aware of the missing density for K^{16.50} when inspecting Figure 5D, the respective figure legend now reads:

“Close-up view and comparison of the semi-conserved glutamate (E^{6.53/57}) orientation and helix 5 conformation in *Pa*PPase:Mg₅IDP and *Vr*PPase:Mg₅IDP. Dashed lines shown the coordination of key residues involved in ion selectivity. K^{16.50} is not defined in the electron density but modelled in the stereochemically most plausible pose (Appendix Table S6), which aligns with its orientation in *Vr*PPase:Mg₅IDP. Major structural changes are indicated by black arrows”

Comment 9: - line 359-360: I do not understand the reasoning. Hydrolysis is assumed to be slow, yet the protein pumps to ions when PP is added, and hydrolysis of 1 PP is needed to get both ions pumped. Is the rate of that first hydrolysis event fast? Or slow? Different from the second event?

We are aware that this sections is difficult to understand for readers outside the M-PPase field. We have thus tried to rephrase it to make it more accessible, especially in the context of other changes made (see above). This section now reads:

“We further used electrometric measurements to study energy-coupling of PP_i hydrolysis and Na⁺ pumping in *Tm*PPase. In this experiment, currents are generated when ions cross the membrane, so the measured current is the sum of the currents from all active proteins on the sensor. In the presence of 200 μ M K₂HPO₄ as a negative control, the current was about 0.025 nA (Figure 8A): phosphate can bind in the active site as shown in structure *Tm*PPase:Mg₄P₁₂ (PDB: 4AV6) and *Vr*PPase:Mg₂P_i (PDB: 5GPI) but does not cause ion pumping. A positive signal of 0.24 \pm 0.005 nA was detected after addition of 100 μ M substrate (K₄PP_i) and reached its maximum within \sim 0.1 seconds, *i.e.* in the dead time of the machine under the conditions used (Figure 8). When the substrate was

replaced by 100 μM of the non-hydrolysable analogue IDP, the signal was reduced by about half to $0.09 \pm 0.009 \text{ nA}$. This is at least 12-25 times faster than the hydrolysis rates at 20 °C under similar lipidated conditions (Appendix Fig S5C, Appendix Table S7). The simplest explanation of these data is that the electrometric signals obtained by IDP correspond to a single pumping event, whereas PP_i triggers two pumping events. This also means that the overall turn-over is rate limited by PP_i hydrolysis or phosphate release. Indeed, we observed that to fully recover the signal on the same sensor, a waiting time of several minutes was required, in line with the proposition that hydrolysis or phosphate release is slow. The signal decayed within about 2 seconds, corresponding to *TmPPase* entering a state that temporarily could no longer pump Na^+ at a sufficient rate to generate a signal. The current decay curves (Figure 8 B-C) were well fit by a single exponential with similar decay rates (k) for PP_i (2.1 s^{-1}) and IDP (3.4 s^{-1}) (Table 3). Overall, these measurements are consistent with *TmPPase* pumping Na^+ upon addition of either PP_i or IDP (see Discussion), with PP_i generating two pumped Na^+ (one from binding the first PP_i and one from binding the second one after hydrolysis of the first) and IDP one (as no hydrolysis can take place)."

Our point is that pumping of the first ion is fast (as shown by electrometry) but that hydrolysis (and/or product release) is slow. Our expectation, but we have no data to prove this, is that pumping of the second ion would also be fast, once substrate is bound to the second subunit. But substrate binding to the second subunit is slow and limited by the rate-determining-step at the first subunit, which is most likely hydrolysis. This is visualised in Figure 9.

Dear Prof. Goldman

Thank you for the submission of your revised manuscript to EMBO reports. I have already informed you about the referee reports we received and I have now completed all checks from the editorial side. I am sorry about the delay but after my conference travel, I could not complete it as fast as I had wished.

All referees are very positive about the study and request only minor changes to clarify methods and conclusions. From the editorial side, there are also a few things that we need before we can proceed with the official acceptance of your study.

- Please update the 'Conflict of interest' paragraph to our new 'Disclosure and competing interests statement'. For more information see

<https://www.embopress.org/page/journal/14693178/authorguide#conflictsofinterest>

- Please remove the Author Contributions from the manuscript file and make sure that the author contributions in our online submission system are correct and up-to-date. The information you specified in the system will be automatically retrieved and typeset into the article. You can enter additional information in the free text box provided, if you wish.

- Reference format: et al needs to be used after 10 author names the DOIs need to be removed. You can download the respective EndNote file from our Guide to Authors, if you wish:

https://endnote.com/style_download/embo-reports/

- Author Checklist: Please complete the information on corresponding author name, ID# and journal in the rosy boxes at the top-left corner.

- The funding information needs to be part of the Acknowledgments. The information on funding in the manuscript needs to match that specified in the online submission system. In that respect we note that the information on Cluster of Excellence "The Hamburg Centre for Ultrafast Imaging" and "CUI: Advanced Imaging of Matter" is missing.

- Please remove the legends from the separately uploaded figures.

- Please add individual callouts for Figure 7 (A and B) in the manuscript text where appropriate.

- Will the files "Maps and coordinates of structures" and "PDB validation reports of structures" remain part of the published manuscript? I assume these files were for the reviewers only and are part of the PDB submissions and the source data. If you keep any of these files, please upload them with e.g. the filetype Dataset EV#.

- The legends for the two EV tables should be removed from the manuscript as they are already provided in the individual files.

- Appendix: please provide page numbers in the file and in the table of content.

- The movie needs to be called out as Movie EV1 in the manuscript text. It needs to be zipped with its legend which can be provided as a README.txt file.

- The Data availability section should only refer to data deposited in external repositories. Therefore, please remove the following statement: All data needed to evaluate the conclusions in the paper are presented here. Additional data related to this paper may be requested from the authors.

- Please shorten the title to 100 characters including spaces.

- Can you please upload the source data separately for each figure (one zipped folder per one figure)?

- Finally, EMBO Reports papers are accompanied online by A) a short (1-2 sentences) summary of the findings and their significance, B) 2-3 bullet points highlighting key results and C) a synopsis image that is 550x300-600 pixels large (width x height) in PNG for JPG format. You can either show a model or key data in the synopsis image. Please note that the size is rather small and that text needs to be readable at the final size. Please send us this information along with the revised manuscript.

- On a different note, I would like to alert you that EMBO Press offers a new format for a video-synopsis of work published with us, which essentially is a short, author-generated film explaining the core findings in hand drawings, and, as we believe, can be very useful to increase visibility of the work. This has proven to offer a nice opportunity for exposure i.p. for the first author(s) of the study.

Please see the following link for representative examples and their integration into the article web page:

<https://www.embopress.org/doi/full/10.15252/emboj.2019103932>

With kind regards,

Referee #1:

My concerns from a previously submitted manuscript to EMBO Journal have been addressed beautifully. The arguments presented now fit the data presented and the analytical level of the paper is excellent. I fully support the publication of this paper.

Referee #2:

Thank you for addressing the concerns about the structural analysis and SSM assay in the updated manuscript.

I am still confusing about the Nanion assay. Here I paste my question and your reply.

Comment 3: In the SSM assay, the authors used 100 micromolar of PPI and 50 micromolar IDP to activate the assay. Without information on the K_m of these substrates, how do the authors conclude: "TmPPase Pump two Na^+ with PPI and 1 Na^+ with IDP.

IDP is not a substrate in mPPases, but a competitive inhibitor so a K_m cannot be determined. A full catalytic cycle, which would in all models involve pumping from both active sites, i.e. 2 Na^+ , can therefore not be achieved with IDP as it is not hydrolysed, a prerequisite for getting the second subunit ready for pumping. In other words, ion and substrate binding to subunit B cannot take place until hydrolysis happens in subunit A - and no hydrolysis can happen with IDP. The key fact is that PPI leads to the expected pumping activity, IDP leads to some pumping activity, but the very close isostere etidronate does not, indicating a significant difference in their behaviour. We showed this for *Vigna radiata* (mung bean) mPPase in 2016 (Li et al., Nat Commun 2016; 7: 13596). This is explained in the following sentences:

"The measured currents in presence of IDP are specifically caused by a single ion pumping event and not by Mg^{2+} being brought slightly beneath the membrane surface as PPI mimics that allow loop5-6 closure and thus pumping (e.g. IDP) induce a bigger signal than those that do not (e.g. etidronate), and these signals can be collapsed by H^+ -specific ionophores (Li et al., 2016, Shah et al., 2017)."

Nevertheless, we would like to thank the reviewer for pointing out that the use of 100 micromolar PPI, but 50 micromolar IDP weakens our data interpretation. To ensure that the differences in SSM results are not due to a lower affinity of IDP for TmPPase, we have repeated the experiment with 100 micromolar IDP and updated the respective figure panel and methods section. We observed no significant difference to the previously collected data, which unambiguously shows that at {greater than or equal to}50 micromolar IDP binding is already saturated and that our interpretation of the data holds. We believe that the result of the additional experiment thus satisfies the referees concern.

I hope my understanding of this assay is correct. So, in the assay, you have sodium-free buffer inside the liposome. A non-activated sodium buffer follows through the sensor, and the sodium will bind to the protein. This event should happen before the 1 second, not shown in the figure. I am OK with that.

According to Figure 9, the complete cycle of Na^+ pumping is generated when PPI is hydrolyzed to two Pi. My question is, dose the binding of PPI and followed hydrolysis of PPI contribute to the current in the black trace?

It shows the assay records both of the bounding IDP and the release of the bounded Na^+/H^+ in the top right corner of Figure 9. Considering the charge of IDP is -4, and the binding of IDP needs the binding of 5 Mg^{2+} , the binding event is electrogenic. The

total charge in the binding event is +6. As IDP is not hydrolyzed, how Na⁺ is pumped? I am wondering what you measure in the blue trace is the binding of the IDP-Mg complex. It is not Na⁺ pumping.

Minor:

Please update the figure legend of Figures 1 and 10. The sodium is shown as a purple sphere, not blue.

Referee #3:

The authors have adequately addressed my previous concerns.

Point-by-point response letter EMBO R submission: EMBOR-2023-58049-T

Comment 1: I hope my understanding of this assay is correct. So, in the assay, you have sodium-free buffer inside the liposome. A non-activated sodium buffer follows through the sensor, and the sodium will bind to the protein. This event should happen before the 1 second, not shown in the figure. I am OK with that.

According to Figure 9, the complete cycle of Na^+ pumping is generated when PP_i is hydrolyzed to two P_i . My question is, does the binding of PP_i and followed hydrolysis of PP_i contribute to the current in the black trace?

It shows the assay records both of the bounding IDP and the release of the bounded Na^+/H^+ in the top right corner of Figure 9. Considering the charge of IDP is -4, and the binding of IDP needs the binding of 5 Mg^{2+} , the binding event is electrogenic. The total charge in the binding event is +6. As IDP is not hydrolyzed, how Na^+ is pumped? I am wondering what you measure in the blue trace is the binding of the IDP-Mg complex. It is not Na^+ pumping.

We would like to apologize for not having made our discussion of the electrometric data clear enough and reviewed the respective parts to ensure a better understanding (see next paragraph). The electrometric assay measures currents when ions cross the membrane, *i.e.* ion pumping. This is described in line 358. Furthermore, we would like to point out that substrate does not enter as $\text{Mg}_5\text{PP}_i/\text{Mg}_5\text{IDP}$ complex. The number of paired Mg^{2+} ions depend in a complex way on the pH and Mg concentration. Under assay conditions, PP_i (and IDP) most likely bind as Mg_2PP_i complex (Maeshima, BBA Biomembranes 2000; 8711465:1-2, Malinen et al., Biochemistry 2008; 87847:50). The remaining Mg^{2+} ions of the metal cage that coordinate substrate in an active enzyme state (and are mapped in the structures) are already present at the active site of M-PPases prior to the arrival of substrate. This is described in line 80. To make this clearer, we have added following sentence to the discussion of the electrometric data in line 436 (highlighted in yellow):

“Moreover, PP_i (or IDP) most likely enter as Mg_2 -complex under assay conditions (Maeshima, 2000; Malinen et al., 2008), which is electro neutral and thus cannot trigger a signal. Consequently, pumping must precede hydrolysis.”

To improve the structuring and readability of the manuscript and reduce its complexity, we have relocated the two following parts from the electrometric results to the discussion section and removed redundant statements.

“The simplest explanation of these data is that the electrometric signals obtained by IDP correspond to a single pumping event, whereas PP_i triggers two pumping events. This also means that the overall turn-over is rate limited by PP_i hydrolysis or phosphate release.”

“Overall, these measurements are consistent with TmPPase pumping Na^+ upon addition of either PP_i or IDP (see Discussion), with PP_i generating two pumped Na^+ (one from binding the first PP_i and one from binding the second one after hydrolysis of the first) and IDP one (as no hydrolysis can take place).”

We similarly restructured parts of the discussion to better group content about half-of-the-sites reactivity and energy-coupling together, the content itself was not changed. Sections that were rearranged for a better understanding are highlighted in blue. The restructured discussion about energy-coupling that is largely based on electrometric data now reads:

“We continue with the controversy discussed mechanism of energy coupling in M-PPase. *Tm*PPase catalysis as measured by phosphate production has a k_{cat} of 0.2-0.4 s⁻¹ at 20 °C (Appendix Table S7) but maximal signal in the electrometric measurements is reached in the dead time of the machine (~0.1 seconds), and the height of the peak for PP_i is about twice that of IDP (Figure 8). Consequently, the rate of ion pumping is at least 12-25 times that of PP_i hydrolysis/phosphate release. The decay constants (protein unable to pump) for the two are similar (Table 3), suggesting that they correspond to a similar event. We thus hypothesize that PP_i hydrolysis/phosphate release from the liposomes at 20 °C is rate limiting. Indeed, we observed that to fully recover the signal on the same sensor, a waiting time of several minutes was required, in line with the proposition that hydrolysis or phosphate release is slow. The logical explanation for the twice as big signal from PP_i as compared to IDP is that IDP leads to a single pumping event, whereas PP_i triggers two pumping events. In other words, with PP_i the enzyme can bind and pump two Na⁺ within a catalytic cycle (one from each subunit, with at least a single turnover in subunit A) while non-hydrolysable IDP only allows the binding and pumping of one Na⁺ (Figure 9). In the context of half-of-the-sites reactivity, this means that ion and substrate binding to subunit B cannot take place until hydrolysis happens in subunit A – and no hydrolysis can happen with IDP. Similar results have been reported for a K⁺-dependent H⁺-PPase in which PP_i produced a ~10-fold bigger signal (multiple turnovers) than other non-hydrolysable PP_i mimics (single-pumping event upon binding of one molecule) (Li et al., 2016, Shah et al., 2017). The measured currents in presence of IDP are specifically caused by a single ion pumping event and not by Mg²⁺ being brought slightly beneath or close to the membrane surface as PP_i mimics that allow loop₅₋₆ closure and thus pumping (e.g. IDP) induce a bigger signal than those that do not (e.g. etidronate), and these signals can be collapsed by ionophores specific for the pumped ion (Li et al., 2016, Shah et al., 2017). Moreover, PP_i (or IDP) most likely enter as Mg₂-complex under assay conditions (Maeshima, 2000; Malinen et al., 2008), which is electro neutral and thus cannot trigger a signal. Consequently, pumping must precede hydrolysis.”

We believe that these changes make it much easier to follow our conclusions and answer all additional questions regarding the cause of the observed signals in our electrometric assay that were raised by referee 2.

Comment 1: As IDP is not hydrolyzed, how Na⁺ is pumped?

Since IDP cannot be hydrolysed but triggers an ion-pumping event, energy coupling must follow a pumping-before-hydrolysis mechanism. Following this mechanism, the arrival of PP_i, or close homologues that allow proper close of the active site such as IDP but not etidronate, induce structural changes that cause ion-pumping (Li et al., Nat Commun 2016;7: 13596). In a full catalytic cycle (PP_i as substrate), each subunit of the homo dimeric M-PPase pumps one ion across the membrane (Figure 9). If PP_i is replaced by IDP, the completion of the full catalytic cycle is inhibited as IDP cannot be hydrolysed (Figure 9), which is a prerequisite for setting up the second subunit for binding (half-of-the-sites-reactivity) (Artukka et al., Biochem J 2018; 475:6, Vidilaseris et al., Sci Advances 2019; 5:5). Consequently, only one ion can be pumped. This aligns nicely with IDP inducing half the signal of PP_i in the electrometric assay. We believe that the changes made to the respective discussion section (see above) explain this now more clearly and fully answer the question raised by referee 2.

Comment 1: It shows the assay records both of the bounding IDP and the release of the bounded Na⁺/H⁺ in the top right corner of Figure 9.

As pointed out earlier and in line 358, the electrometric assay only measures ions crossing the membrane. Figure 9 shows a schematic of our proposed model of M-PPase catalysis that combines half-of-the-sites reactivity with pumping-before-hydrolysis. Red boxes simply highlight enzyme states mapped by time-resolved crystallography *or* investigated in electrometric studies as described in the figure legend. We are aware that Figure 9 shows a complex mechanism, which is why we decided to update it slightly to depict the pumping and hydrolysis event in two individual steps rather than in a single step to clarify that pumping takes place before hydrolysis according to our proposed model of catalysis. This should also help with the understanding of the IDP-inhibited state as the newly added enzyme conformation in Figure 9 now shows an equivalent state with PP_i being bound.

Comment 2: Please update the figure legend of Figures 1 and 10. The sodium is shown as a purple sphere, not blue.

We thank the reviewer for spotting this mislabelling and have implemented the suggested changes.

Prof. Adrian Goldman
University of Helsinki
Molecular and Integrative Biosciences
Helsinki 00100
Finland

Dear Prof. Goldman,

I am very pleased to accept your manuscript for publication in the next available issue of EMBO reports. Thank you for your contribution to our journal.

Yours sincerely,
